# NPY-mediated synaptic plasticity in the extended amygdala prioritizes feeding during starvation

Stephan Dodt[1,2], Noah V. Widdershooven[2], Marie-Luise Dreisow[1,3], Lisa Weiher[1], Lukas Steuernagel [2], F. Thomas Wunderlich [2,3,4,5], Jens C. Brüning [2,3,4,5] ✉ & Henning Fenselau [1,3,4] ✉

Efficient control of feeding behavior requires the coordinated adjustment of complex motivational and affective neurocircuits. Neuropeptides from energy-sensing hypothalamic neurons are potent feeding modulators, but how these endogenous signals shape relevant circuits remains unclear. Here, we examine how the orexigenic neuropeptide Y (NPY) adapts GABAergic inputs to the bed nucleus of the stria terminalis (BNST). We find that fasting increases synaptic connectivity between agouti-related peptide (AgRP)-expressing 'hunger' and BNST neurons, a circuit that promotes feeding. In contrast, GABAergic input from the central amygdala (CeA), an extended amygdala circuit that decreases feeding, is reduced. Activating NPY-expressing AgRP neurons evokes these synaptic adaptations, which are absent in NPY-deficient mice. Moreover, fasting diminishes the ability of CeA projections in the BNST to suppress food intake, and NPY-deficient mice fail to decrease anxiety in order to promote feeding. Thus, AgRP neurons drive input-specific synaptic plasticity, enabling a selective shift in hunger and anxiety signaling during starvation through NPY.

An organism's ability to tightly tune motivational systems is critical for promoting food intake during states of energy deprivation. A key motivational feature that promotes the acquisition and consumption of food is the facilitation of hunger drive. In addition, motivational systems that detract from, or even eliminate, feeding must be suppressed. This includes the reduction of fear and anxiety in order to increase risk-taking and foraging behavior in environments where obtaining food is difficult or even life-threating. To coordinate this trade-off balance in motivational drives, the activity and dynamics of the underlying neural circuits must be appropriately adapted so that food seeking and food consumption dominate over other motivated behaviors during states of starvation[1–4].

A large body of literature suggests that neuropeptides are key coordinators of such motivational circuitry tuning[5–7]. Within this broad class of neuroactive chemicals, pharmacological, knock-out, as well as cell type-specific manipulation studies have implicated neuropeptide Y (NPY) as the one most strongly associated with both increasing hunger drive and decreasing anxiety[5,8–15]. Importantly, the hunger-promoting effects of NPY have been linked to the starvation-induced activation of Agouti-related peptide (AgRP) neurons of the arcuate nucleus (ARC)[13,14], which show the highest expression of NPY in the hypothalamus[16,17]. Mice lacking NPY show a diminished increase in food intake following fasting and depletion of NPY abolishes the rapid as well as prolonged increases in feeding upon selective AgRP neuron

[1]Synaptic Transmission in Energy Homeostasis Group, Max Planck Institute for Metabolism Research, Gleueler Strasse 50, 50931 Cologne, Germany. [2]Department of Neuronal Control of Metabolism, Max Planck Institute for Metabolism Research, Gleueler Strasse 50, 50931 Cologne, Germany. [3]Center for Endocrinology, Diabetes and Preventive Medicine (CEDP), University Hospital Cologne, Kerpener Strasse 26, 50924 Cologne, Germany. [4]Excellence Cluster on Cellular Stress Responses in Aging Associated Diseases (CECAD), University of Cologne, Joseph-Stelzmann-Straße 26, Cologne 50931, Germany. [5]Center of Molecular Medicine Cologne (CMMC), University of Cologne, Robert-Koch-Straße 21, 50931 Cologne, Germany. ✉ e-mail: bruening@sf.mpg.de; henning.fenselau@sf.mpg.de

stimulation[13,14,18]. Further, NPY is a potent suppressor of fear[19], and AgRP neuron activation decreases anxiety-related behavior and increases risk behavior to maximize food acquisition[20–25]. Together, these findings imply that AgRP neurons function as a central control point to sense caloric deficit and, in turn, to coordinate dynamic adaptations of hunger and anxiety through NPY release. While this model provides a compelling explanation for how feeding behavior can be efficiently orchestrated by this discrete neuronal population, it leaves unanswered the question of which motivational and affective neural circuits are subject to NPY-mediated tuning.

AgRP neurons drive behavioral adaptations through their projections to multiple, separate brain regions[4,26–29]. One prominent candidate region for tuning the balance between hunger and anxiety is the bed nucleus of the stria terminals (BNST). AgRP neurons send dense projections to the BNST, and optogenetic stimulation of this circuit elicits intense feeding within minutes[26,30,31]. Further, AgRP neuron projections to the BNST are permissive for evoking the anxiolytic effects of fasting as well as for stimulating food acquisition and consumption under threat of predation[24,28]. Moreover, the BNST is thought to constitute an essential part of the neurocircuitry that controls anxiety-related behavior, particularly through its integration

of fear-related signals emanating from the central amygdala (CeA). Indeed, opto- and chemogenetic manipulations have demonstrated that the CeA→BNST circuit is necessary and sufficient for controlling behavioral adaptations to threat exposure[32–35]. Despite the potential implications for both hunger drive and anxiety signaling, the effects of caloric restriction on these synaptic inputs to BNST neurons remain unexplored. It is also unknown whether there is any NPY-mediated plasticity of AgRP→BNST and CeA→BNST synapses.

In the present study, we combined circuit-specific electrophysiological and optogenetic approaches with chemogenetic manipulations in transgenic mice to determine the contribution of NPY to synaptic adaptations of the AgRP→BNST and CeA→BNST circuit in feeding behavior regulation. We focused on GABAergic synaptic transmission, since GABA is the sole fast-acting neurotransmitter that mediates transmission at AgRP neuron synapses[17,36–38] and also relays fear-related signals between CeA and BNST neurons[32,34,39]. Our experiments revealed that the activation of starvation-sensing AgRP neurons drives input-specific forms of plasticity at these two distinct GABAergic afferents onto BNST neurons, and that NPY is uniquely required for these synaptic effects. Further, we found that the ability of the CeA→BNST circuit to suppress feeding is diminished upon fasting,

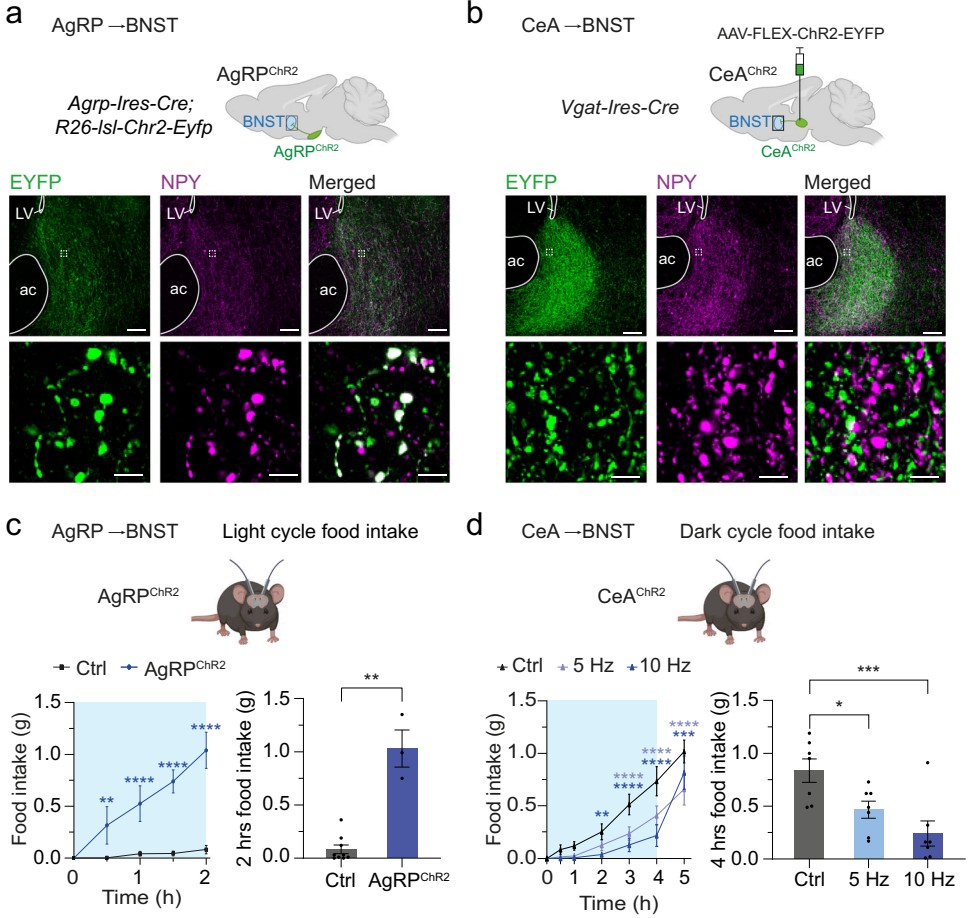

**Fig. 1 | Control of feeding behavior via distinct GABAergic inputs to the BNST.**
**a**, **b** Schematic illustration of the approach used for optogenetic stimulation of AgRP neuron terminals in the BNST of mice expressing ChR2 exclusively in AgRP neurons (AgRP^ChR2 mice) (**a**) and axonal projections in the BNST of mice expressing ChR2 in GABAergic neurons of the CeA (CeA^ChR2 mice) (**b**). Representative images showing ChR2-EYFP (green) and NPY (magenta) expression in the BNST in AgRP^ChR2 mice (**a**) and CeA^ChR2 mice (**b**). **c** Cumulative and total light cycle food intake during photostimulation (20 Hz; 1 s on, 3 s off) of AgRP^ChR2 neuron terminals in the BNST compared to mice without expression of ChR2 (Ctrl; N = 9/3, Ctrl/AgRP^ChR2 animals). **p = 0.0091, ****p < 0.0001 (two-way ANOVA with Šidák post hoc test, left),

**p = 0.0045 (one-sided Mann–Whitney test, right). Blue box indicates time of photostimulation. **d** Cumulative and total dark cycle food intake during photostimulation (5 Hz or 10 Hz) of CeA^ChR2 projections in the BNST compared to no photostimulation (Ctrl; N = 7 animals). **p = 0.0017, ***p = 0.0007, ****p < 0.0001 (two-way ANOVA with Šidák post hoc test, left), *p = 0.0212, ***p = 0.0008 (one-way ANOVA with Šidák post hoc test, right). Scale bars: 100 μm (**a**, **b**; top), 5 μm (**a**, **b**; bottom); ac anterior commissure, LV lateral ventricle. All data are presented as mean ± SEM. Asterisks indicate significant differences to the control condition. Schematics in (**a**–**d**) were created with Biorender.com released under a Creative Commons Attribution-NonCommercial-NoDerivs 4.0 International license.

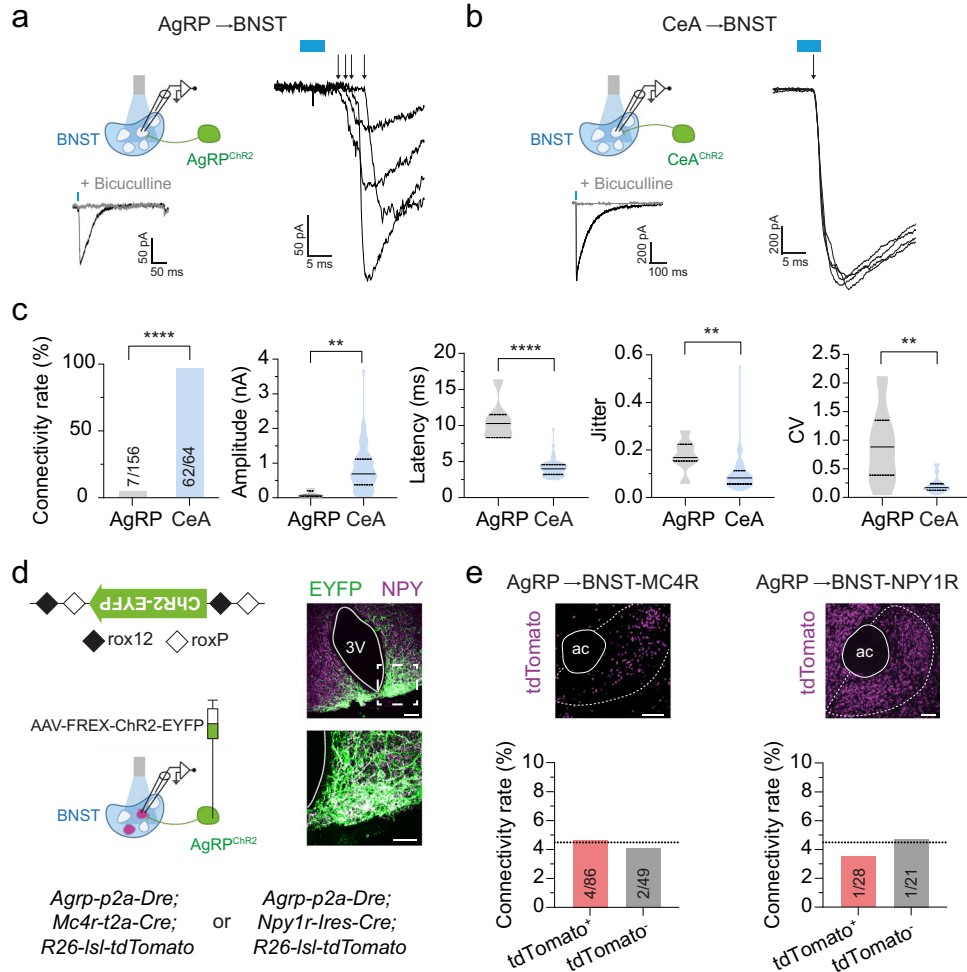

**Fig. 2 | GABAergic AgRP→BNST and CeA→BNST synapses have distinct characteristics. a, b** Schematic illustration of the approach used for electrophysiological characterization of GABAergic transmission across AgRP→BNST synapses in AgRP[ChR2] mice (**a**) and CeA→BNST synapses in CeA[ChR2] mice (**b**). Representative traces from voltage-clamp recordings of light-evoked inhibitory postsynaptic currents (eIPSCs) recorded from randomly selected neurons in the ChR2-expressing projection fields in the BNST. eIPSCs were completely blocked following bath application of the GABA$_A$ receptor antagonist bicuculline. Arrows indicate eIPSC onsets. Blue boxes indicate light pulses. **c** Summary of connectivity rates for the AgRP→BNST circuit and the CeA→BNST circuit. Summaries of amplitudes, latencies, jitter, and coefficient of variation (CV) of eIPSCs (17/7 brain slices from 13/5 AgRP[ChR2]/CeA[ChR2] animals). ****$p < 0.0001$ (two-sided Fisher's exact test), **$p = 0.0044$ (two-sided unpaired $t$-test), ****$p < 0.0001$, **$p = 0.002$, **$p = 0.0038$ (two-sided Mann–Whitney test). **d** Schematic illustration of the AAV-FREX-ChR2-EYFP construct and the approach used for determining GABAergic synaptic

connectivity between AgRP neurons and BNST-MC4R or BNST-NPY1R neurons (tdTomato-expressing). Representative images showing expression of the Dre-dependent ChR2-EYFP (green) and NPY (magenta) in the ARC. **e** Representative images showing tdTomato expression in the BNST of *Mc4r-t2a-Cre; R26-lsl-tdTomato* and *Npy1r-Ires-Cre; R26-lsl-tdTomato* mice as determined by fluorescence in situ hybridization. Summaries of connectivity rates between AgRP neurons and BNST-MC4R neurons (tdTomato[+]; 10 brain slices from 7 animals) or BNST-NPY1R neurons (tdTomato[+]; 5 brain slices from 3 animals). Dashed lines indicate the connectivity rate to randomly selected BNST neurons in AgRP[ChR2] mice (**c**). Scale bars: 100 μm (**d**, top; **e**); 5 μm (**d**, bottom); ac, anterior commissure; 3 V, third ventricle. Numbers in bars indicate BNST neurons with eIPSCs in relation to all recorded neurons. Bar graphs represent average connectivity rates (**c**, **e**). Violin plots represent median ± quartiles (**c**). Schematics in (**a**, **b**, and **d**) were created with Biorender.com released under a Creative Commons Attribution-NonCommercial-NoDerivs 4.0 International license.

and that NPY deficiency renders mice unable to increase food acquisition and consumption in an anxiogenic environment. The homo- and heterosynaptic tuning of discrete GABAergic inputs to the BNST reported here provides a mechanistic basis for the effective adaptation of feeding behavior under caloric starvation, and suggests a novel role for AgRP neuron-derived NPY in gating motivational systems through synaptic plasticity.

## Results
### Distinct GABAergic inputs to the BNST differently control feeding behavior

To compare and contrast the input-specific characteristics by which GABAergic projections to the BNST from AgRP and CeA neurons contribute to feeding regulation, we induced cell type-specific

expression of the optogenetic activator Channelrhodopsin-2 (ChR2) in these two distinct afferent neuronal populations. For the assessment of AgRP neuron input, we crossed *Agrp-Ires-Cre* mice with *R26-lsl-Chr2-Eyfp* mice for the expression of ChR2 tagged to enhanced yellow fluorescent protein (EYFP) in AgRP neurons (AgRP[ChR2]; Fig. 1a and Supplementary Fig. 2a). To assess GABAergic input emanating from the CeA, we bilaterally injected adeno-associated viruses (AAV) expressing Cre-dependent ChR2 (AAV-FLEX-ChR2-EYFP or -mCherry) into the CeA of *Vgat-Ires-Cre* mice (CeA[ChR2]; Fig. 1b and Supplementary Fig. 1b, 2a). Optic fibers were bilaterally implanted above the BNST (Supplementary Fig. 2a). Immunohistochemical analysis of ChR2-linked fluorophores confirmed that both AgRP[ChR2] and CeA[ChR2] neurons send strong projections to dorsal and ventral subregions of the BNST, but not to the nearby nucleus accumbens (Fig. 1a, b, and

Supplementary Fig. 1a, b). Importantly, we found that the projection fields of both afferent neuronal populations widely overlapped (Fig. 1b), indicating that AgRP and CeA neurons engage similar, or even the same, downstream neurons in the BNST.

As previously observed[26,30,31], photostimulation of AgRP[ChR2] neuron terminals in the BNST rapidly and profoundly increased food intake in sated mice at the onset of the light cycle (Fig. 1c and Supplementary Fig. 2b). 2 h of photostimulation (20 Hz; 1 s on, 3 s off) caused an overall food intake of ~1 g of chow diet in AgRP[ChR2] mice, but not in control mice lacking ChR2 expression in AgRP neurons (Ctrl; Fig. 1c). In contrast, photostimulation of CeA[ChR2] terminals in the BNST potently decreased food intake at the onset of the dark cycle, when mice are naturally hungry (Fig. 1d and Supplementary Fig. 2c). Specifically, food intake was reduced by continuous photostimulation with a frequency of 5 Hz and higher, a stimulation protocol that was found to evoke anxiety and feeding behavior[40,41], while stimulating at 2.5 Hz had no effect (Fig. 1d and Supplementary Fig. 2c), demonstrating stimulus intensity-dependent regulation of feeding suppression by the CeA→BNST circuit. Notably, when the photostimulation was switched off after 4 h, mice rapidly increased food intake (Fig. 1d and Supplementary Fig. 2c), indicating that continuous and steady activation of this GABAergic circuit is of importance for feeding reduction. Photostimulation of CeA[ChR2] terminals in the BNST also caused a slight reduction in food intake of sated mice during the light cycle (Supplementary Fig. 2d). Of note, continuous photostimulation of AgRP→BNST projections with a frequency of 5 Hz also promoted food intake in sated AgRP[ChR2] mice, yet to a lesser extent (Supplementary Fig. 2b). The optogenetic stimulation protocol did not affect food intake in control mice, which lacked ChR2 expression (Supplementary Fig. 2e).

## GABAergic AgRP→BNST and CeA→BNST synapses have distinct characteristics

The strikingly opposing changes in feeding that we observed upon stimulating BNST projections in AgRP[ChR2] and CeA[ChR2] mice suggest that GABAergic AgRP→BNST and CeA→BNST inputs have mechanistically distinct features through which they control the activity of downstream BNST neurons. To explore this possibility, we employed an optogenetic-electrophysiology approach. We prepared acute brain slices from AgRP[ChR2] and CeA[ChR2] mice and performed whole-cell patch clamp recordings from randomly selected BNST neurons in the projection fields (Fig. 2a, b). Recordings were made in voltage-clamp configuration ($V_h = -70$ mV) with a CsCl-based internal solution. Light illumination (473 nm wavelength, 5 ms) evoked time-locked inhibitory postsynaptic currents (eIPSCs) in AgRP[ChR2] and CeA[ChR2] mice. These eIPSCs were completely blocked by bath application of the GABA_A receptor antagonist bicuculline (Fig. 2a, b), confirming the GABAergic nature of both inputs.

Analysis of the recordings revealed profound differences in transmission across AgRP→BNST and CeA→BNST synapses. Specifically, connectivity rates in AgRP[ChR2] mice were remarkably low, with eIPSCs detected in only about 4.5% of BNST neurons (7 out of 156 cells; Fig. 2c). In contrast, in CeA[ChR2] mice, eIPSCs were detected in virtually every BNST neuron (62 out of 64 cells; Fig. 2c). Additional analysis of eIPSC induction per given light pulse revealed that the response rate was lower for the AgRP→BNST circuit than for the CeA→BNST circuit (Supplementary Fig. 3a). Further, amplitudes of eIPSCs in AgRP[ChR2] mice were smaller, whereas eIPSC latencies were larger (Fig. 2c).

To determine whether these differences in synaptic properties were due to variances in the release probability of neurotransmitters from axonal terminals, we determined the jitter, the coefficient of variability (CV), the paired pulse (PP) ratio, and the PP probability of eIPSCs, which constitute complementary parameters for the characterization of presynaptic function[42,43]. We found remarkable differences between the AgRP→BNST and the CeA→BNST circuit in all four parameters; the jitter, CV and PP ratio were higher for the AgRP→BNST

circuit than for the CeA→BNST circuit, whereas the PP probability was lower (Fig. 2c and Supplementary Fig. 3a, b). Together, these findings demonstrate that GABAergic transmission across the AgRP→BNST synapse is weak and unreliable, whereas that across the CeA→BNST synapse is robust and potent. This is, at least in part, due to differences in their release probability.

The low connectivity rate between AgRP neurons and randomly selected BNST neurons was surprising given that previous studies have reported high connectivity rates between AgRP neurons and neurons in other projection sites that evoke similar feeding responses—for example the paraventricular hypothalamus (PVH), where we and others have detected light-evoked currents in ~30–40% of randomly selected neurons[13,14,38,44]. To reexamine this, we assessed the connectivity rate of the AgRP→PVH circuit in AgRP[ChR2] mice (Supplementary Fig. 3c). Consistent with previous reports[13,14,38,44], we detected eIPSCs in ~35% (14 out of 39) of PVH neurons (Supplementary Fig. 3c). We also determined whether virally-mediated expression of ChR2 impacts the connectivity rate of the AgRP→BNST circuit. We injected AAV-FLEX-ChR2-EYFP into the ARC of Agrp-Ires-Cre mice (AgRP[AAV-ChR2]; Supplementary Fig. 3d). We found that the connectivity rate of the AgRP→BNST circuit in AgRP[AAV-ChR2] mice was similarly low as with our transgenic approach (3 out of 70 cells) (Supplementary Fig. 3d).

Given the low connectivity rate, we explored whether AgRP neurons preferentially engage melanocortin 4 receptor (MC4R)- or neuropeptide 1 receptor (NPY1R)-expressing neurons in the BNST. This question is of interest because AgRP neurons have been shown to form GABAergic synaptic connections with neurons expressing these receptors for the AgRP neuron-derived neuropeptides (i.e., AgRP and NPY, respectively) with a particular high preference – although in other projection sites[2,4,30,45]. We assessed GABAergic synaptic connectivity between AgRP neurons and MC4R-BNST or NPY1R-BNST neurons by employing a Dre- and Cre-recombinase-utilizing approach (Fig. 2d, e). AgRP-p2a-Dre mice were crossed with Mc4r-t2a-Cre; R26-lsl-tdTomato or Npy1r-Cre; R26-lsl-tdTomato mice (Fig. 2d). The resulting triple transgenic mice were injected with an AAV expressing Dre-dependent ChR2-EYFP (AAV-FREX-ChR2-EYFP, see methods) into the ARC (Fig. 2d). We confirmed selective and efficient ChR2 expression in AgRP neurons in the ARC and their terminals in the BNST (Fig. 2d and Supplementary Fig. 3e, f). Analysis of the connectivity rate using this targeted mapping approach revealed that MC4R- and NPY1R-BNST neurons (tdTomato[+]) are not preferentially engaged by AgRP neurons (Fig. 2e). Thus, unlike in other brain regions - such as the PVH - AgRP neurons do not provide selective GABAergic input to BNST neuron subtypes that express receptors for the neuropeptides they release.

## Fasting evokes input-specific forms of plasticity in the BNST

Energy deprivation activates AgRP neurons, and this increases hunger drive to promote food consumption[38,46,47]. While the release of GABA, NPY, and AgRP from axonal terminals has been linked to AgRP neuron-mediated stimulation of feeding behavior[13,14,38,48,49], it remains poorly understood how these inhibitory signals shape neural communication in downstream projection sites. To begin to explore the underlying mechanisms, we first determined how caloric deprivation impacts transmission across the AgRP→BNST synapse. To this end, AgRP[ChR2] mice were sacrificed for electrophysiological recordings following an overnight fast (Fig. 3a). We found that fasting profoundly increased the probability of detecting eIPSCs in randomly selected BNST neurons to ~12% (20 out of 167 cells; Fig. 3b and Supplementary Fig. 4a). In addition, we found that fasting decreased the average amplitude of eIPSCs (from 0.11 nA to 0.05 nA; Fig. 3b). Parameters indicative for changes in release probability of GABA (jitter, CV, PP ratio, and PP probability) did not significantly differ between fed and fasted mice (Fig. 3c and Supplementary Fig. 4b). The overall increase in synaptic connectivity together with the reduction in eIPSCs amplitude and the absence of changes in presynaptic function indicates that fasting evokes the

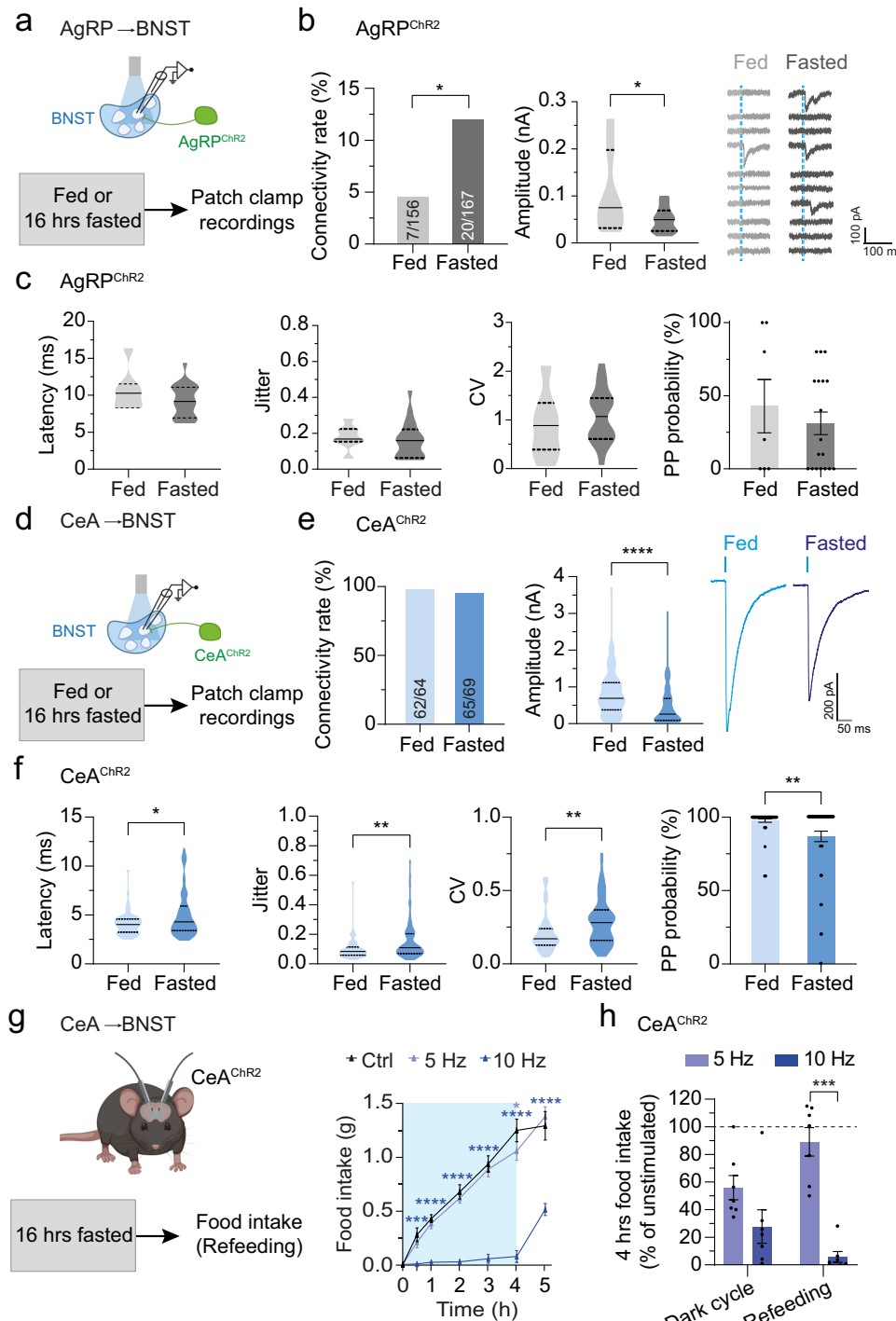

formation of new functional synaptic connections or 'unsilencing' of GABAergic AgRP→BNST neuron synapses.

We next determined fasting-induced alterations in transmission across GABAergic CeA→BNST synapses (Fig. 3d). Fasting decreased the average amplitudes of eIPSCs in BNST neurons of CeA[ChR2] mice (from 0.84 nA to 0.48 nA) whereas the connectivity rate of this circuit remained unchanged (Fig. 3e). Further analysis demonstrated that fasting increased the latencies, jitter, and CV of eIPSCs (Fig. 3f), suggesting a reduction in presynaptic neurotransmitter release probability. Consistent with this, fasting augmented the PP probability (Fig. 3f) and the response rate to light stimulation in CeA[ChR2] mice, while no change in the PP ratio was found (Supplementary Fig. 4c).

Together, these findings demonstrate that energy deprivation evokes strikingly different alterations in GABAergic synaptic input on BNST neurons emanating from AgRP and CeA neurons; transmission across AgRP→BNST synapses is amplified whereas that across CeA→BNST synapses is diminished.

### Fasting reduces the behavioral impact of the GABAergic CeA→BNST circuit

The reduced transmission across CeA→BNST GABAergic synapses upon fasting suggests that the ability of the circuit to suppress feeding is diminished during starvation. To test this possibility, we assessed the effects of optogenetically stimulating GABAergic CeA→BNST projections on food intake in fasted CeA[ChR2] mice (Fig. 3g). Consistent with

**Fig. 3 | Energy deprivation evokes input-specific forms of plasticity at AgRP→BNST and CeA→BNST synapses. a** Illustration of the approach used for characterizing AgRP→BNST synapses. **b** Summary of connectivity rates for the AgRP→BNST circuit and eIPSC amplitudes in the BNST of AgRP[ChR2] mice (16/14 brain slices from 13/11 fed/fasted animals). Representative traces illustrating the ratio of neurons with synaptic input from AgRP neurons under fed/fasted conditions. Dashed blue lines indicate light pulses. *$p = 0.0162$ (two-sided Fisher's exact test), *$p = 0.0191$ (two-sided unpaired $t$-test). **c** Summaries of latencies, jitter, CV, and PP probabilities of eIPSCs in randomly selected BNST neurons of AgRP[ChR2] mice (16/14 brain slices from 13/11 fed/fasted animals). **d** Illustration of the approach used for characterizing CeA→BNST synapses. **e** Summaries of connectivity rates for the CeA→BNST circuit and eIPSC amplitudes in BNST neurons of CeA[ChR2] mice (6/6 brain slices from 5/5 fed/fasted animals). ****$p < 0.0001$ (one-sided Mann–Whitney test). **f** Summaries of latencies, jitter, CV, and PP probabilities of eIPSCs in BNST neurons of CeA[ChR2] mice (6/6 brain slices from 5/5 fed/fasted animals). *$p = 0.0354$,

**$p = 0.0034$, **$p = 0.0011$, **$p = 0.0018$ (one-sided Mann–Whitney test). **g** Illustration of the approach used for in vivo optogenetic stimulation of GABAergic CeA[ChR2] terminals in the BNST in fasted mice. Cumulative food intake during photostimulation (5 Hz or 10 Hz) compared to no stimulation (Ctrl; $N = 7$ animals). *$p = 0.0127$, ***$p = 0.0003$; ****$p < 0.0001$ (two-way ANOVA with Šidák post hoc test). Blue box indicates time of photostimulation. **h** Relative change in dark cycle food intake and post-fast refeeding during photostimulation (5 Hz or 10 Hz) of GABAergic CeA[ChR2] terminals in the BNST (Ctrl; $N = 7$ animals). ***$p = 0.0009$ (two-way ANOVA with Šidák post hoc test). Dashed line: 100%, without photostimulation. Numbers in bars indicate BNST neurons with eIPSCs in relation to all recorded neurons. Bar graphs represent average connectivity rates (**b**, **e**) or mean ± SEM (**c**, **f**, **h**). Violin plots represent median ± quartiles (**b**, **c**, **e**, and **f**). Asterisks indicate significant differences to the control condition. Schematics in (**a**, **d**, and **g**) were created with Biorender.com released under a Creative Commons Attribution-NonCommercial-NoDerivs 4.0 International license.

the profound feeding suppression we observed at the onset of the dark cycle (Fig. 1d), photostimulation of CeA[ChR2] terminals in the BNST with a frequency of 10 Hz decreased food intake in fasted mice (Fig. 3g, h). By contrast, photostimulation with a frequency of 5 Hz, which decreased dark cycle feeding (Fig. 1d), was completely ineffective in reducing food intake in fasted mice (Fig. 3g, h). Thus, the fasting-induced reduction of GABAergic transmission in the CeA→BNST circuit is associated with a reduction in its anorexigenic potency. In agreement with our observations during the dark cycle, food intake rapidly increased when the photostimulation was switched off in the refeeding paradigm (Fig. 3g).

## NPY is required for the fasting-evoked adaptations of GABAergic AgRP→BNST and CeA→BNST synapses

Given the critical importance of NPY in promoting food intake upon activation of starvation-sensing AgRP neurons[13,14], we next probed the involvement of this orexigenic neuropeptide in the fasting-induced synaptic adaptations in the BNST. First, we determined how starvation affects the amount of NPY in the BNST. We found that NPY expression was increased in fasted mice as determined by fluorescence immunohistochemistry (Fig. 4a and Supplementary Fig. 5a). Of note, our analysis also revealed that the vast majority of NPY-expressing terminals originate from AgRP neurons, both under fed and fasted conditions (Fig. 4b and Supplementary Fig. 5b), indicating that the fasting-induced increase in NPY expression stems from AgRP neurons. Next, we assessed the necessity of NPY in driving the fasting-induced adaptations in transmission across AgRP→BNST synapses. We crossed NPY knockout (NPY-KO) mice[50] with AgRP[ChR2] mice to generate NPY-deficient mice that express ChR2-EYFP exclusively in AgRP neurons (NPY-KO::AgRP[ChR2] mice). We confirmed that NPY-KO::AgRP[ChR2] mice lack NPY as determined by immunohistochemistry (Fig. 4b and Supplementary Fig. 5c, d). Electrophysiological recordings from randomly selected BNST neurons showed that NPY-KO::AgRP[ChR2] mice exhibit no differences in transmission across AgRP→BNST synapses under fed conditions (Fig. 4c-e and Supplementary Fig. 5e). Importantly, fasting failed to increase the connectivity rate of the AgRP→BNST circuit in NPY-KO::AgRP[ChR2] mice as compared to control NPY-WT::AgRP[ChR2] mice (Fig. 4d). Thus, NPY expression in the BNST is upregulated during starvation and is required for increasing GABAergic transmission across AgRP→BNST synapses.

Next, we probed the necessity of NPY in the fasting-induced attenuation of transmission across CeA→BNST synapses. We virally expressed ChR2 in the CeA of NPY-KO mice for the generation of NPY-KO::CeA[ChR2] mice (Fig. 4f). Importantly, the fasting-induced adaptations in GABAergic transmission were completely abolished, or even reversed, in NPY-KO::CeA[ChR2] mice (Fig. 4g, h, and Supplementary Fig. 5f). Specifically, we found that fasting increased average eIPSC amplitudes recorded from BNST neurons in NPY-KO::CeA[ChR2] mice (from 0.68 nA – 1.04 nA; Fig. 4g), whereas eIPSC's CV was reduced as

compared to fed NPY-KO::CeA[ChR2] mice (from 0.25 to 0.18; Fig. 4h). All other electrophysiological parameters remained unchanged upon fasting (Fig. 4g, h, and Supplementary Fig. 5f). Thus, as with AgRP→BNST synapses, NPY is uniquely required for the fasting-induced adaptations of CeA→BNST synapses.

## Lack of NPY abolishes increases in food acquisition and consumption in an anxiogenic environment

Previous studies found that AgRP[24,28] and CeA[32–35] projections to the BNST regulate context-dependent adaptations of anxiety-related behaviors. Based on our findings demonstrating the necessity of NPY in triggering synaptic plasticity in both circuits upon fasting, we next probed how NPY deficiency affects energy state-dependent behavioral changes in the elevated O-maze (EOM) test, which allows to determine anxiety-related behaviors[51,52]. We adapted a protocol in which objects or food pellets were placed in the middle zones of the open arms[20] (Fig. 5a). Consistent with the capability of caloric restriction to decrease anxiety-like behavior[20,21,23], fasted NPY wildtype (NPY-WT) mice spent considerably more time in the open arms (Fig. 5b, c). This increase was paralleled by an enhancement in food acquisition and food consumption (Fig. 5c and Supplementary Fig. 6a). Although fasted NPY-WT mice displayed no changes in the overall amount of running distance, they remained longer in the closed arm before entering the open area (latency; Supplementary Fig. 6a), presumably due to a delayed initiation of behavioral output to preserve energy. In contrast, when we assessed the behavior of NPY-KO mice in the EOM, we found that the fasting-induced increases in time spent in the open arms and food acquisition were completely abolished (Fig. 5d and Supplementary Fig. 6b). Moreover, fasted NPY-KO mice showed no increase in food consumption in the EOM (Fig. 5d), demonstrating that NPY action is required for increasing foraging behavior to obtain and consume food in an anxiogenic environment during starvation. Of note, during home cage refeeding, food intake within the first 20 min after returning food did not significantly differ between NPY-WT and NPY-KO mice (Supplementary Fig. 6c), while food intake was significantly lower in NPY-KO mice after 60 min of refeeding (Supplementary Fig. 6c). In addition, photostimulation of AgRP[ChR2] terminals in the BNST caused a significantly weaker feeding response in NPY-KO mice (Supplementary Fig. 6d). Thus, the specific reduction of feeding behavior in the EOM is consistent with our model that the fasting-induced decrease in anxiety signaling is due to AgRP neuron-derived NPY.

To further investigate the role of AgRP neuron-derived NPY to adjustments in anxiety-related behaviors via the BNST, we bilaterally implanted optic fibers above the BNST in NPY-WT and NPY-KO mice that express ChR2 in AgRP neurons (Fig. 5e). Optogenetic stimulation of AgRP[ChR2] terminals in the BNST (ON) of NPY-WT mice resulted in an increase in time spent on the open arms when compared to no stimulation (OFF; Fig. 5f). In contrast, there were no significant behavioral changes in NPY-KO mice (Fig. 5g). This further supports our hypothesis

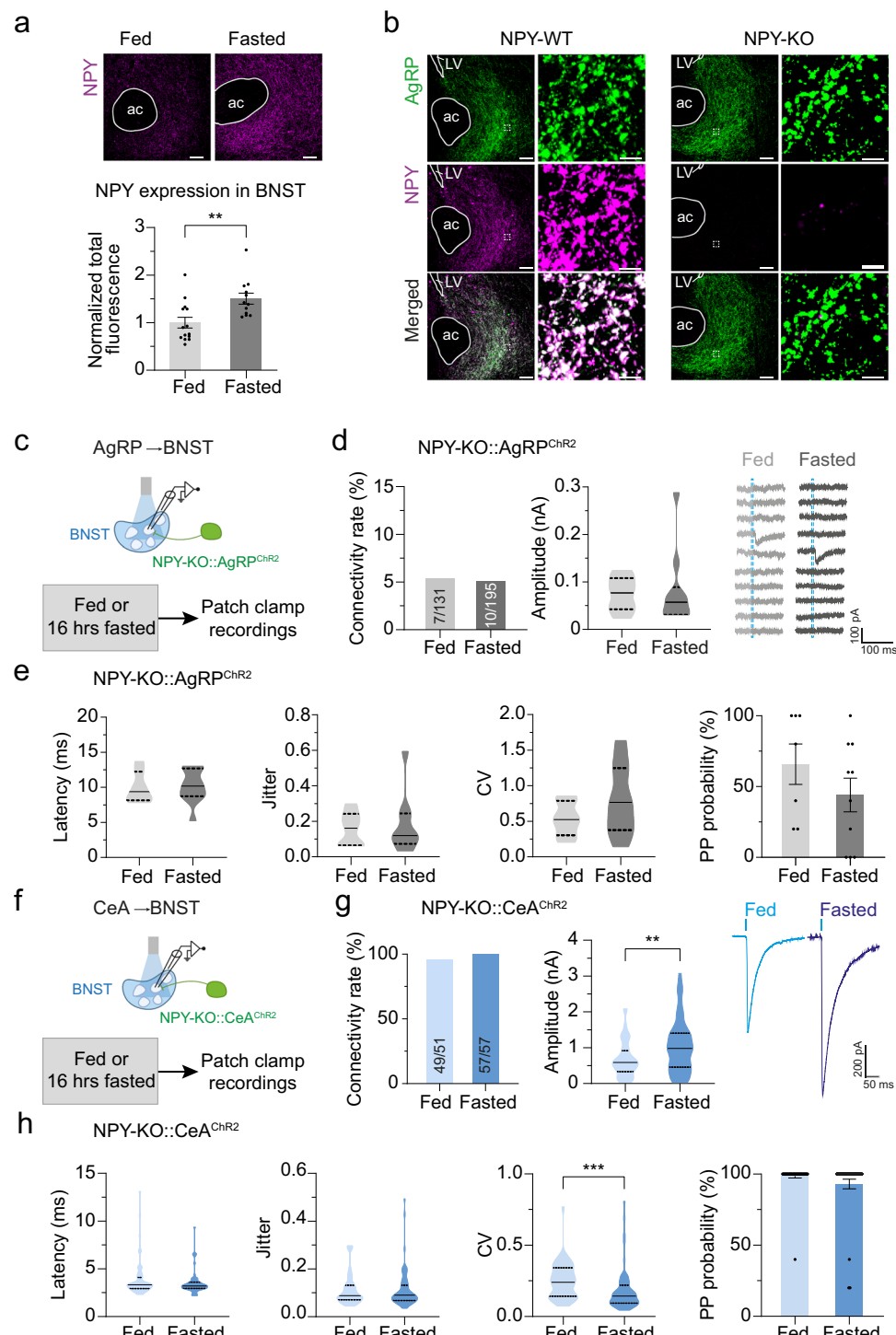

that increased NPY release from activated AgRP neurons adapts anxiety-related behaviors via the BNST.

## Acute activation of AgRP neurons is sufficient for evoking the NPY-dependent synaptic adaptations in the BNST

NPY is found in the BNST, CeA and AgRP neurons[16,17,53–58]. Given the key role of NPY from AgRP neurons in adjusting feeding behavior[13,14,59], we next probed whether AgRP neuron activation, and concomitant release of NPY from their terminals, orchestrates the fasting-induced adaptation of GABAergic synaptic inputs to BNST neurons. We virally expressed mCherry (Ctrl) or the chemogenetic activating receptor hM3Dq in AgRP neurons of AgRP[ChR2] and CeA[ChR2] mice that express NPY

(NPY-WT::AgRP[ChR2] and NPY-WT::CeA[ChR2]) as well as in NPY-deficient mice (NPY-KO::AgRP[ChR2] and NPY-KO::CeA[ChR2]; Fig. 6a). We confirmed efficient activation of AgRP neurons following administration of the hM3Dq actuator clozapine N-oxide (CNO); hM3Dq/CNO-induced activation of AgRP neurons promoted robust food intake in NPY-WT mice—as has been reported previously[60] (Fig. 6b and Supplementary Fig. 7a). To avoid any confounding effects of altered food intake on GABAergic afferents of BNST neurons following CNO administration, mice did not have access to food until brain slices were prepared for electrophysiological recordings (Fig. 6a). When we assessed AgRP→BNST and CeA→BNST inputs, we found that AgRP neuron activation evoked the same synaptic adaptations of GABAergic

**Fig. 4 | NPY deficiency abrogates the fasting-induced synaptic adaptations of GABAergic inputs to the BNST. a** Representative images showing NPY expression (magenta) in the BNST of a fed/fasted wildtype mouse. Intensity of NPY antibody fluorescence in the BNST ($n = 14/12$ brain sections sampled from 3/3 fed/fasted animals). **$p = 0.0049$ (two-sided unpaired $t$-test). **b** Representative images of the BNST showing almost complete overlap of NPY (magenta) and AgRP (green) expression in a wildtype (NPY-WT) mouse, and lack of NPY expression in AgRP neuron terminals of an NPY-deficient (NPY-KO) mouse. **c** Schematic illustration of the approach used for electrophysiological characterization of the AgRP→BNST circuit in fed/fasted NPY-KO::AgRP$^{ChR2}$ mice. **d** Summaries of connectivity rates for the AgRP→BNST circuit and eIPSC amplitudes in BNST neurons of NPY-KO::AgRP$^{ChR2}$ mice (16/17 brain slices from 12/13 fed/fasted animals). Representative traces from voltage-clamp recordings of BNST neurons illustrating the ratio of neurons with synaptic input from AgRP neurons under fed/fasted conditions. Dashed blue lines indicate light pulses. **e** Summaries of latencies, jitter, CV, and PP probabilities of eIPSCs in BNST neurons of NPY-KO::AgRP$^{ChR2}$ mice (16/17 brain slices from 12/13 fed/

fasted animals). **f** Schematic illustration of the approach used for electrophysiological characterization of the CeA→BNST circuit in fed/fasted NPY-KO::CeA$^{ChR2}$ mice. **g** Summaries of connectivity rates for the CeA→BNST circuit and eIPSC amplitudes in BNST neurons of NPY-KO::CeA$^{ChR2}$ mice (3/4 brain slices from 3/4 fed/fasted animals). Representative traces from voltage-clamp recordings of eIPSCs recorded from BNST neurons in NPY-KO::CeA$^{ChR2}$ mice. **$p = 0.0093$ (two-sided Mann–Whitney test). **h** Summaries of latencies, jitter, CV, and PP probabilities of eIPSCs in BNST neurons of NPY-KO::CeA$^{ChR2}$ mice (3/4 brain slices from 3/4 fed/fasted animals). ***$p = 0.0002$ (two-sided Mann–Whitney test). Scale bars: 100 μm (**a**, **b**), 5 μm (**b**); ac anterior commissure, LV lateral ventricle. Numbers in bars indicate BNST neurons with eIPSCs in relation to all recorded neurons. Bar graphs represent average connectivity rates (**d**, **g**) or mean ± SEM (**a**, **e**, **h**). Violin plots represent median ± quartiles (**d**, **e**, **g**, **h**). Schematics in (**c**) and (**f**) were created with Biorender.com released under a Creative Commons Attribution-NonCommercial-NoDerivs 4.0 International license.

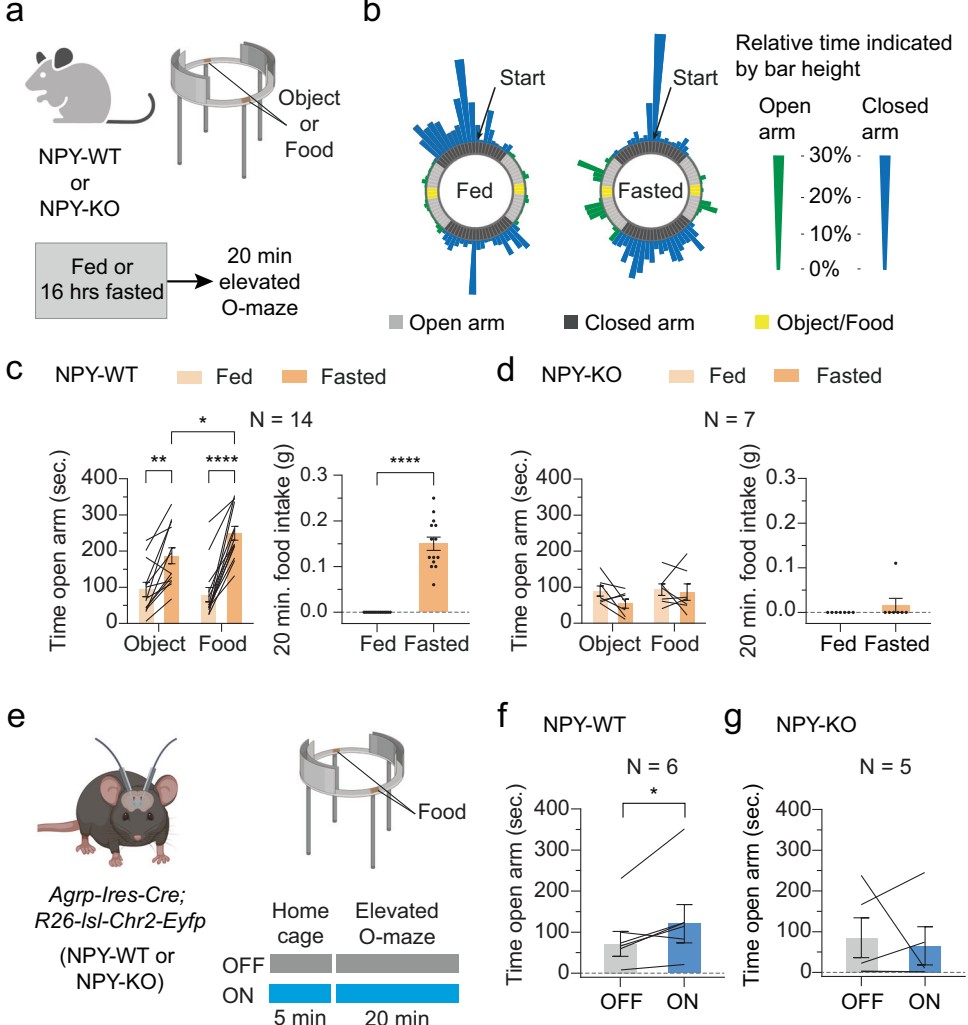

**Fig. 5 | NPY deficiency abolishes adjustments in feeding behavior in an anxiogenic environment. a** Schematic illustration of the approach used for the investigation of anxiety-related behaviors of wildtype (NPY-WT) and NPY-deficient (NPY-KO) mice in the elevated O-maze (EOM) under fed/fasted conditions.
**b** Representative heat maps of a fed/fasted NPY-WT mouse in the EOM. Bar height indicates relative time spent in the segment of the EOM. Blue/green bars represent relative time spent in the closed/open arms. **c**, **d** Summaries of time spent on the open arms and overall food intake on the open arms of the EOM of fed/fasted NPY-WT (**c**, $N = 14$ animals) and NPY-KO mice (**d**, $N = 7$ animals). *$p = 0.0434$, **$p = 0.0035$,

****$p < 0.0001$ (two-way ANOVA with Šidák post hoc test), ****$p < 0.0001$ (one-sided Wilcoxon test). **e** Schematic illustration of the approach used for the investigation of anxiety-related behaviors of NPY-WT/NPY-KO mice in the EOM upon photostimulation of AgRP$^{ChR2}$ terminals in the BNST. **f**, **g** Summaries of time spent on the open arms of the EOM of NPY-WT (**f**, $N = 6$ animals) and NPY-KO mice (**g**, $N = 5$ animals) with/without (ON/OFF) photostimulation. *$p = 0.0469$ (one-sided Wilcoxon test). All data are presented as mean ± SEM. Schematics in (**a**) and (**e**) were created with Biorender.com released under a Creative Commons Attribution-NonCommercial-NoDerivs 4.0 International license.

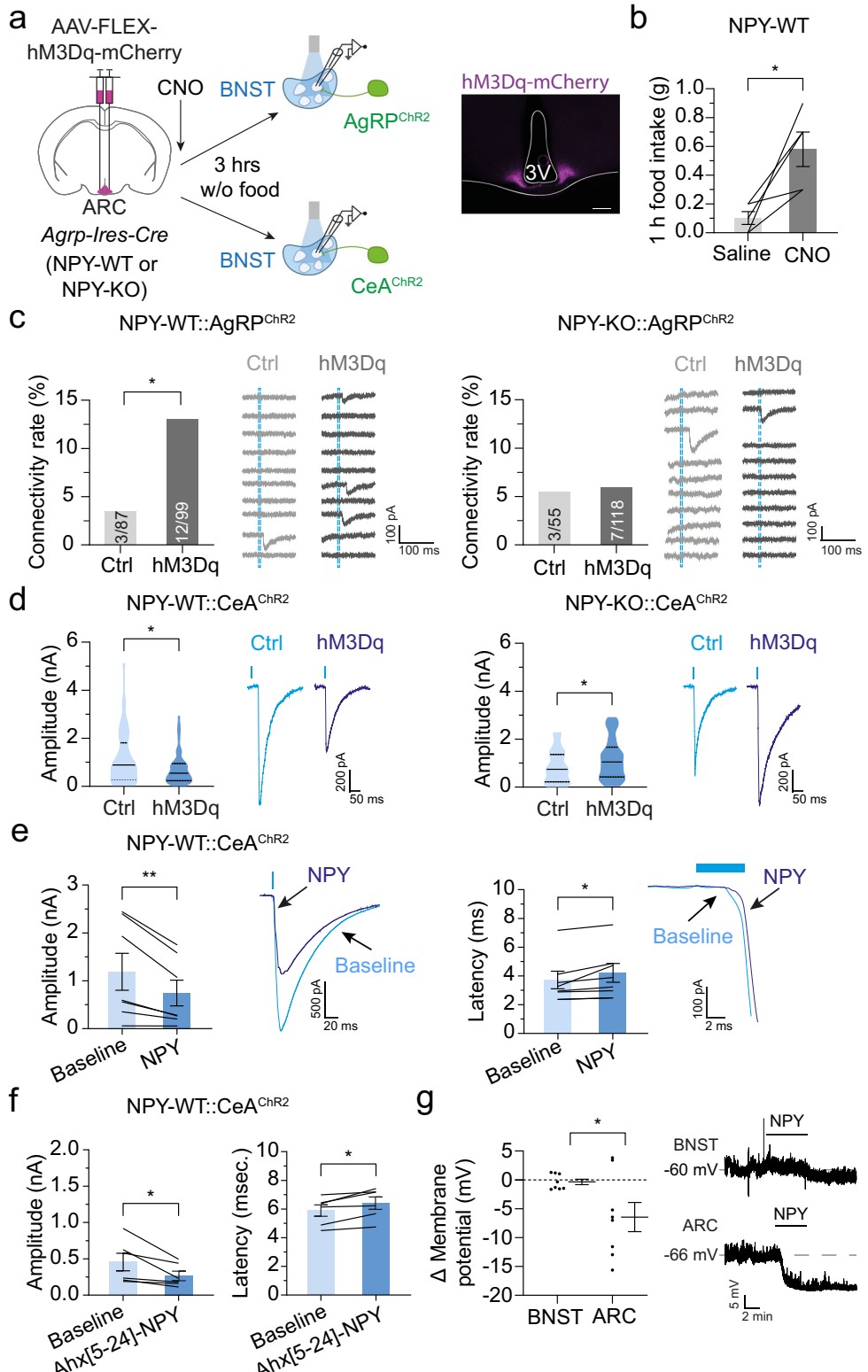

transmission as observed following fasting (Fig. 6c, d, and Supplementary Fig. 7b–e). Moreover, as for the fasting-evoked adaptations in both circuits, NPY was required for evoking AgRP neuron-induced synaptic changes (Fig. 6c, d, and Supplementary Fig. 7c, e). Specifically, we found that AgRP neuron activation in NPY-WT::AgRP[ChR2] mice increased the percentage of BNST neurons with detectable eIPSCs (3.4% vs 12.1%; Fig. 6c) while other electrophysiological parameters remained unchanged (Supplementary Fig. 7b). This increase in

AgRP→BNST connectivity following acute AgRP neuron activation was absent in NPY-KO::AgRP[ChR2] mice (Fig. 6c). In addition, we found that AgRP neuron activation in NPY-WT::CeA[ChR2] mice reduced the amplitudes of eIPSCs recorded from BNST neurons (Fig. 6d), while eIPSC latencies, jitter, and CV were increased (Supplementary Fig. 7d). In contrast, AgRP neuron activation in NPY-KO::CeA[ChR2] mice increased eIPSC amplitudes (Fig. 6d), while latencies, jitter, and CV were reduced (Supplementary Fig. 7e). Thus, AgRP neuron activation, which occurs

**Fig. 6 | NPY is required for the homo- and heterosynaptic adaptations evoked by AgRP neuron activation. a** Approach for chemogenetic activation of AgRP neurons in NPY-WT and NPY-KO AgRP^ChR2/CeA^ChR2 mice. Representative image showing hM3Dq-mCherry expression in the ARC. **b** Food intake of NPY-WT mice expressing hM3Dq in AgRP neurons following i.p. injection of saline/CNO during light cycle (N = 5 animals). *p = 0.024 (two-sided paired t-test). **c** Summaries of connectivity rates for the AgRP→BNST circuit in NPY-WT::AgRP^ChR2/NPY-KO::AgRP^ChR2 mice with/without (hM3Dq/Ctrl) hM3Dq expression. (7/7 brain slices from 5/5 NPY-WT::AgRP^ChR2 animals (Ctrl/hM3Dq); 5/9 brain slices from 4/7 NPY-KO::AgRP^ChR2 animals (Ctrl/hM3Dq)). Representative traces from voltage-clamp recordings illustrating the ratio of BNST neurons with synaptic AgRP neuron input. Dashed blue lines indicate light pulses. *p = 0.0251 (one-sided Fisher's exact test). **d** Summaries of eIPSC amplitudes and representative traces from voltage-clamp recordings of BNST neurons in NPY-WT::CeA^ChR2/NPY-KO::CeA^ChR2 mice with/without hM3Dq expression (4/5 brain slices from 4/4 NPY-WT::CeA^ChR2 animals (Ctrl/hM3Dq); 3/4 brain slices from 2/4 NPY-KO::CeA^ChR2 animals (Ctrl/hM3Dq)). *p = 0.0346 (one-sided Mann–Whitney test), *p = 0.0497 (one-sided unpaired t-test). **e** Summaries of eIPSC amplitudes/latencies and

representative traces from voltage-clamp recordings of BNST neurons in NPY-WT::CeA^ChR2 mice before/after (Baseline/NPY) bath application of NPY (0.3 μM; n = 7 cells). **p = 0.0078 (one-sided Wilcoxon test), *p = 0.018 (one-sided paired t-test). **f** Summaries of eIPSC amplitudes/latencies and representative traces from recordings of BNST neurons in NPY-WT::CeA^ChR2 mice before/after (Baseline/Ahx[5-24]-NPY) bath application of the NPY2R agonist Ahx[5-24]-NPY (1 μM; n = 6 cells). *p = 0.0219, *p = 0.0109 (one-sided paired t-test). **g** Summary of changes in membrane potentials and representative traces from current-clamp recordings illustrating the effect of NPY on the membrane potential of NPY1R-expressing neurons in the BNST/ARC following bath application of NPY (0.3 μM; both n = 8 cells). *p = 0.0327 (two-sided unpaired t-test). Scale bar: 100 μm (**a**); 3 V, third ventricle. Numbers in bars indicate BNST neurons with eIPSCs in relation to all recorded neurons. Bar graphs and scatter plots represent mean ± SEM (**b**, **e**, **f**, **g**) or average connectivity rates (**c**). Violin plots represent median ± quartiles (**d**). Schematic in (**a**) was created with Biorender.com released under a Creative Commons Attribution-NonCommercial-NoDerivs 4.0 International license.

during caloric restriction, is sufficient to evoke the input-specific forms of synaptic adaptation in the BNST through NPY.

The observed changes at CeA→BNST synapses suggest that AgRP-neuron derived NPY is capable of evoking heterosynaptic adaptations. To further define NPY's ability in regulating the CeA→BNST circuit, we assessed the acute effects of NPY on eIPSCs in NPY-WT::CeA^ChR2 mice (Fig. 6e). Addition of NPY (0.3 μM) decreased eIPSC amplitudes in BNST neurons while eIPSC latencies were increased (Fig. 6e). Since projections of GABAergic neurons emanating from the CeA abundantly express the NPY2R at their axonal terminals in the BNST[56], we hypothesized that this receptor might be a possible candidate for mediating the neuromodulatory effects of NPY. Consistent with this prediction, bath application of the selective NPY2R-agonists Ahx[5-24]-NPY (1 μM) or NPY13-36 (1 μM) caused comparable changes in eIPSC amplitudes and latencies in NPY-WT::CeA^ChR2 mice (Fig. 6f and Supplementary Fig. 7f). Together, these findings suggest that NPY2R activation is, at least in part, responsible for the AgRP neuron-mediated suppression of GABAergic afferents emanating from the CeA, and supports the concept of a heterosynaptic inhibitory mechanism on presynaptic terminals. Given that virtually all BNST neurons showed GABAergic input from the CeA, we also determined the effects of NPY and NPY13-36 on spontaneous GABA release. Consistent with previous studies[61,62], bath application of either compound evoked a robust reduction in the frequency, but not amplitude, of spontaneous IPSCs (sIPSCs) (Supplementary Fig. 7g, h).

Our results thus far demonstrate that NPY coordinates the strength of GABAergic synaptic inputs for the selective tuning of feeding signals. To test whether NPY also controls BNST neuron activity through postsynaptic mechanisms, we determined how NPY addition affects their membrane potential. To specifically focus on BNST neurons whose membrane potential could be regulated by NPY, we studied NPY1R-expressing neurons by recording from tdTomato-expressing cells in *Npy1r-Cre*; *R26-lsl-tdTomato* mice (Fig. 6g). We found that NPY addition failed to affect the membrane potential of NPY1R-expressing neurons located in the BNST (Fig. 6g). Of note, control recordings from NPY1R-expressing neurons in the ARC showed that NPY caused a pronounced hyperpolarization (Fig. 6g) - as has been previously observed[63,64]. This suggests that NPY does not regulate BNST neuron activity via postsynaptic mechanisms.

## Discussion
To promote food seeking and consumption during states of caloric deficit, hunger drive must override competing motivational systems. Recent work has shown that AgRP neurons play a key role in the reduction of competing incentives[1,2,20], in addition to their undisputable function in promoting appetite during caloric deficit[47]. However,

how the increased release of the inhibitory neurotransmitters and neuropeptides from AgRP neurons shape activity of relevant downstream circuits remains largely unclear. The selective innervation patterns, combined with findings from projection-specific manipulation studies[2,4,24,26–29,65], suggest that AgRP neurons precisely shape communication of discrete synaptic connections to coordinate multiple physiological and behavioral parameters. If this is the case, it becomes important to define how AgRP neurons organize downstream neural circuits and whether these plastic changes are causally linked to specific physiological processes.

Here, through optogenetic-electrophysiological approaches, we explored two distinct GABAergic afferents of the BNST, a key nucleus of the extended amygdala for the control of anxiety and fear[34,66,67], as well as feeding behavior[26,68]. We focused on the anterior part of the BNST as this region has been shown to receive anxiety-related input from CeA neurons and food intake-regulating input from the hypothalamus[31,69–71]. We found that GABAergic connectivity between AgRP and BNST neurons, which is weak and unreliable under fed conditions, strongly increases upon fasting. In striking contrast, transmission across GABAergic CeA→BNST synapses, which is strong and robust in fed mice, diminishes with fasting (Supplementary Fig. 8). Given that virtually all BNST neurons showed GABAergic input from the CeA, we propose that these input-specific forms of synaptic plasticity co-occur in the same neurons. Importantly, the fasting-induced synaptic changes in both inputs are absent in NPY-deficient mice, raising the likely possibility that NPY's modulatory effects are mediated predominantly through its increased release from activated AgRP neuron terminals, which accounts for most NPY expression in the BNST (Fig. 4b and Supplementary Fig. 5b). Although it is possible that increased levels of NPY could derive from other cells, we demonstrate that selective activation of AgRP neurons is sufficient to induce the changes in GABAergic transmission we observed upon fasting. Moreover, activating AgRP neurons in NPY-deficient mice failed to evoke the fasting-induced patterns of plasticity in both inputs. Given the previously established role of NPY in mediating long-lasting behavioral functions of AgRP neurons, that can be observed even after their inhibition following food acquisition[13,14], the NPY-mediated plasticity reported here provides a synaptic basis for highly selective adjustments of feeding circuits.

As indicated by our comprehensive assessment of electrophysiological parameters, the increase in connectivity of the AgRP→BNST circuit is likely caused by the formation of new synapses. Alternatively, pre-existing 'silent' synapses are recruited to an active state, or the number of functional release sites of existing synapses are increased. Given the above-mentioned finding that AgRP neuron activation is sufficient for increasing connectivity of AgRP→BNST

synapses, we propose that this adaptation is primarily driven by an activity-dependent, homosynaptic mechanism. The NPY-dependent signaling pathways responsible for this GABA synapse strengthening are currently unknown but could involve NPY1Rs or NPY5Rs, which have been found to mediate long-lasting potentiation of GABAergic synapses – although in other brain regions[72–75]. Even though the increase in connectivity across the AgRP→BNST synapse is a plausible mechanism for the promotion of hunger, it is unlikely to be responsible for the anxiolytic phenotype under energy-deprived conditions. This behavioral adaptation is more likely due to the indirect action of AgRP neuron-derived NPY on GABAergic transmission across the CeA→BNST synapse (Supplementary Fig. 8). We propose that, although axonal terminals of both circuits are in close proximity, NPY released from AgRP neurons acts on CeA→BNST projections via volume transmission.

We also demonstrate that NPY is uniquely required to increase food acquisition and food consumption in an anxiogenic environment upon fasting. Several key data provide evidence that this NPY-mediated adjustment of behavior involves synaptic plasticity in the CeA→BNST circuit following AgRP neuron activation. As determined by our histological assessment, as well as previous studies, fasting potently increases NPY in activated AgRP neurons in their projection targets[76], including the BNST (Fig. 4a and Supplementary Fig. 5a). Further, stimulating AgRP neuron projections in the BNST evokes a strong anxiolytic phenotype in fed mice[28], and is sufficient to drive food intake in the presence of a predator[24]. Although stimulating other projection targets of AgRP neurons evokes similar increases in food intake in the home cage environment[26], where animals are exposed to minimal stress, the AgRP neuron-mediated stimulation of feeding behavior under threat was found to be particularly prominent upon activation of BNST projections[24]. Importantly, fasted NPY-deficient mice show little alterations in their acute food intake when monitored in the home cage[13,14] (Supplementary Fig. 6c), indicating the context-specific action of NPY. This, combined with our observation that NPY fails to exert postsynaptic effects on BNST neurons for the control of their excitability, but regulates strength of transmission across AgRP→BNST and CeA→BNST GABAergic synapses, raises the distinct possibility that NPY from AgRP neurons works on GABAergic inputs to produce orexigenic as well as anxiolytic effects.

While previous studies have shown that NPY inhibits GABAergic synaptic input to BNST neurons[61,62], the source of the affected afferents and the recruited NPY receptors were not known. We systematically characterized input-specific forms of synaptic plasticity in this brain region, and identify distinct induction mechanisms that selectively strengthen transmission of one input, but weaken that of the other (Supplementary Fig. 8). As evident from our recordings of light-evoked synaptic currents, addition of NPY or NPY2R agonists reduced the GABAergic input from CeA neurons. The associated increase in eIPSC latency as well as the reduction in sIPSC frequency strongly suggest that a NPY2R-mediated presynaptic mechanism underlies the observed changes (Supplementary Fig. 8); yet, additional postsynaptic effects cannot be excluded. In future histological studies, it will be important to precisely explore the structure and organization of GABAergic inputs to BNST neurons and how NPY shapes them.

Our data demonstrate striking differences in GABAergic transmission of the two synaptic inputs to the BNST as well as opposing energy state-dependent synaptic adaptations. However, it should be considered that we compared a molecularly discrete neuronal population of the hypothalamus (i.e., AgRP neurons) with a likely heterogeneous group of GABAergic neurons of the CeA. The CeA contains multiple molecularly and functionally distinct neuron subtypes, and previous findings demonstrate that discrete neuronal populations project to and synapse on BNST neurons[71]. Thus, further behavioral and mapping studies are required to identify the CeA neuron subtypes that contribute to the energy state-dependent regulation of feeding

behavior through GABA release. The NPY-responsive subtypes are currently unknown, but could correspond to corticotropin-releasing hormone-expressing CeA neurons because their projections to the BNST are necessary and sufficient for the control of anxiety-related behavior[33,34]. Further, while our data from optogenetic stimulation experiments provide strong evidence that fasting-evoked inhibition of GABA release from CeA terminals in the BNST alters feeding behavior, backpropagating action potentials might have evoked the activation of collateral projections to other brain regions. The extent to which such possibility is behaviorally relevant will be an important area for future investigations.

Taken together, we have uncovered a synaptic plasticity mechanism for energy state-dependent tuning of key neurocircuits that control hunger and anxiety signaling. Furthermore, we demonstrate that NPY from activated AgRP neurons is closely linked with coordinating homo- and heterosynaptic adaptations in two distinct circuits and, importantly, finetuning of the motivational drives they control. Viewed in the context of anxiety-related eating disorders, our findings provide circuit-level insights for understanding behavioral maladaptations during states of caloric deficit, which may offer new targeting points for therapeutic strategies.

## Methods
### Animals
All experimental procedures were approved by local government authorities (Bezirksregierung Köln). Mice were monitored for health status daily, housed at 22 – 24 °C on a 12 h light/12 h dark cycle, and had *ad libitum* access to water and to a standard rodent chow diet (ssniff®, V1154) unless food was withdrawn for specific experimental purposes. For all behavioral, histological and electrophysiological studies, male and female adult mice were used.

*Agrp-Ires-Cre*[37] (JAX# 012899), *Vgat-Ires-Cre*[77] (JAX# 016962), *Mc4r-t2a-Cre*[30] (JAX# 030759), *Npy1r-Cre*[2] (JAX# 030544), *R26-lsl-Chr2-Eyfp*[78] (JAX# 012569), *R26-lsl-tdTomato*[79] (JAX# 021876) and *Npy-KO*[50] (JAX# 004545) were previously described and purchased from Jackson Laboratories. *Agrp-p2a-Dre* mice were previously described[80] and kindly provided by Dr. Bradford B. Lowell. *C57Bl/6* mice were purchased from Charles River (Strain code: 027).

All transgenic mice were bred to *C57Bl/6* mice for maintenance. Double transgenic animals and control mice were generated by crossing Cre-expressing mice with Rosa26 transgenic or knockout mice (*AgRP-Ires-Cre; R26-lsl-Chr2-Eyfp, AgRP-Ires-Cre; Npy-KO, Mc4r-t2a-Cre; R26-lsl-tdTomato, Npy1r-Ires-Cre; R26-lsl-tdTomato*). Triple transgenic mice were generated by crossing double transgenic mice with *Agrp-p2a-Dre* or *Npy-KO* mice (*Agrp-p2a-Cre; Mc4r-t2a-Cre; R26-lsl-tdTomato, Agrp-p2a-Cre; Npy1r-t2a-Cre; R26-lsl-tdTomato, AgRP-Ires-Cre; R26-lsl-Chr2-Eyfp; Npy-KO*).

For all experiments, we aimed to include the same number of animals of both sexes. Aside from natural differences, we did not observe any obvious differences in electrophysiological parameters of the investigated neurons, in the expression profile of the investigated neuropeptides, or in the anxiety-related behavior of the mice.

### Stereotaxic surgical procedures
Mice were anesthetized with isoflurane and received an intraperitoneal (i.p.) bolus of Buprenorphine (0.1 mg/kg bodyweight), and were put into a stereotaxic frame (David Kopf Instruments). A local anesthetic agent (Lidocaine) was applied to the skin, the skull surface was exposed through a skin incision, and a small drill hole was made. For virus injections, AAVs (ARC: 200 nl; CeA: 50 nl) were bilaterally delivered through a pulled glass micropipette into the ARC (coordinates from bregma: AP: − 1.5 mm, DV: − 5.95 mm, ML: ± 0.25 mm) and/or into the CeA (coordinates from bregma: AP: −1.35 mm, DV: −4.8 mm, ML: ± 2.35 mm). For fiber placement, two flat tip fiber-optic cannulas (4.5 mm long, 200 µm in core diameter, numerical AP 0.48; Doric

lenses Inc.) were inserted above the BNST (coordinates from Bregma: −1.35 AP, ± 2.35 ML, and −4.8 DV) at an angle of 20° and secured to the skull with dental acrylic (diluted Super Bond C&B). Before waking up, mice received analgesic treatment (subcutaneous injection of Meloxicam (5 mg/kg)) for pain relief and were carefully monitored to ensure regain of pre-surgery weight. All animals were allowed 3–4 weeks for virus expression before starting the experiment. All virus injections and fiber-optic cannula placements were histologically verified after the experiments and mice with missed injections or fiber placements, insufficient expression levels of the virally-mediated transgenes, or expression outside of the target region were excluded from analysis.

## Viruses

AAV-hSyn-DIO-mCherry (#50459-AAV9), AAV-hSyn-ChR2-EYFP (#26973-AAV1), AAV-hSyn-ChR2-mCherry (#26976-AAV8), AAV-EF1α-FLEX-ChR2-EYFP (#20298-AAV1), and AAV-hSyn-DIO-hM3Dq (#44361-AAV9) were purchased from Addgene.

Generation of AAV-CAG-FREX-ChR2-EYFP: A 2323 bp fragment of Ai27 plasmid (Addgene plasmid # 34630) containing ChR2-tdTomato was amplified via PCR using 5 MLU (GAA-TTC) and 3 MLU (GGA-TCC) primers. The PCR product was then subcloned into a pGEM-T Easy vector (Promega, #A3600) containing two rox and rox-mut sites, respectively. Positive clones were digested with EcoRI and BamHI for insolation of the FREX-ChR2-tdTomato insert which was used for ligation into the backbone of a FREX-ZsGreen plasmid (de Solis, unpublished). The resulting construct was digested with AscI in order to release ChR2-tdTomato and was replaced by ChR2-EYFP which was generated by PCR using 5 ascChryfp (TTT-ACG-TCG-CCG-TCC-AGC) and 3 ascChryfp primers (TCA-AGC-CTC-AGA-CAG-TGG-TTC) on the Ai32 plasmid (Addgene plasmid # 34880) template. The resulting construct was termed "AAV-CAG-FREX-ChR2-EYFP" and used for transcranial virus injection for ex vivo electrophysiological experiments.

## Electrophysiology

For electrophysiology, all mice (8–14 weeks of age; males and females) were deeply anesthetized with isoflurane and euthanized by decapitation. Brains were quickly removed into ice-cold cutting solution consisting of (in mM): 92 choline chloride, 30 NaHCO3, 25 Glucose, 20 HEPES, 10 MgSO4, 2.5 KCl, 1.25 NaH2PO4, 5 sodium ascorbate, 3 sodium pyruvate, 2 thiourea, 0.5 CaCl2; oxygenated with 95% O2/5% CO2; measured osmolarity 310–320 mOsm/L. 250 µm thick coronal slices were cut with a Campden vibratome (Model 7000smz-2) (Campden Instruments, Loughborough, UK) and incubated in oxygenated cutting solution at 34 °C for 10 min. Slices were then transferred to oxygenated aCSF consisting of (in mM): 126 NaCl, 21.4 NaHCO3, 2.5 KCl, 1.2 NaH2PO4, 1.2 MgCl2, 2.4 CaCl2, 10 glucose at 34 °C for 45 min., and left to recover in the same solution at room temperature (20–24 °C) for at least 60 min. prior to recordings. Unless otherwise specified, brain slices were prepared 3 h into the light cycle. For patch clamp recordings, a single slice was placed in the recording chamber where it was continuously superfused with oxygenated aCSF at a constant rate of 2–3 ml/min. Recordings were obtained at room temperature from unlabeled neurons or tdTomato-positive neurons in the BNST or ARC visualized with an upright microscope (SliceScope, Scientifica) equipped with a 40x water immersion objective (Olympus, Tokyo, Japan) and a CCD camera (SciCam Pro; Scientifica, Uckfield, UK) as well as infrared differential interference contrast and fluorescence optics.

For whole-cell voltage-clamp recordings, the membrane potential was clamped at Vh = −70 mV and borosilicate patch pipettes (3–5 MΩ) were filled with internal solution consisting of (in mM): 140 CsCl, 2 NaCl, 10 HEPES, 5 EGTA, 2 MgCl2, 0.5 CaCl2, 2 Na2-ATP, 0.5 Na2-GTP, 2 QX 314 bromide (pH 7.3 adjusted with CsOH; 290 mOsm/l). For whole-cell current-clamp recordings, the internal solution contained (in mM): 135 KMeSO3, 10 HEPES, 1 EGTA, 0.1 CaCl2, 4 MgCl2, 4 Na2-ATP, 0.4

Na2-GTP, 5 Na2-phosphocreatine (pH 7.3 adjusted with KOH; 295 mOsm/l). Voltage-clamp recordings were performed in presence of CNQX (10 µM) (Hello Bio, Dunshaughlin, Ireland) and D-AP5 (50 µM) (Alomone labs, Jerusalem, Israel) to block glutamatergic synaptic transmission. Additional application of bicuculline (10 µM) (Sigma-Aldrich, Darmstadt, Germany) was used to verify GABAergic nature of the recorded currents. All recordings were obtained with a Multiclamp 700B amplifier (Molecular Devices, Sunnyvale, USA), Digidata 1550B converter (Axon Instruments, Union City, USA) and pClamp 10.7 software (Molecular Devices, Sunnyvale, USA), sampled at 10 kHz, and filtered at 2 kHz. Access resistance (<30 MΩ) was continuously monitored by a voltage step and recordings were accepted for analysis if changes were <15%.

To photostimulate ChR2-expressing terminals, a LED light source (473 nm) (CoolLED, Andover, UK) was focused onto the back aperture of the microscope objective, producing widefield exposure around recorded cells. The light output was controlled by a programmable pulse stimulator (Master-8; A.M.P.I, Jerusalem, Israel) and pClamp 10.7 software. For recordings of light-evoked inhibitory postsynaptic currents (eIPSC), four blue light pulses (473 nm wavelength, 5 ms) were applied in a 1 s-interval, followed by 6 s without stimulation. For paired-pulse (PP) stimulation, two light pulses were administered 250 ms apart every 15 s.

To investigate the effects of NPY or the NPY2R agonists Ahx[5-24]-NPY and NPY13-36, NPY (0.3 µM) (Tocris, Bristol, UK), Ahx[5-24]-NPY (1 µM) (generous gift from Prof. Dr. Annette G. Beck-Sickinger, Leipzig, Germany), or NPY13-36 (1 µM) (Cayman Chemical Company, Michigan) were added to the aCSF during whole-cell patch clamp recordings. Cells were incubated for at least 10 min. prior to light-stimulation. For current-clamp recordings, following recording the membrane potential at baseline, cells were incubated for at least 5 min. before NPY-free aCSF was again washed into the perfusion system. The membrane potential was continuously recorded.

When recordings were finished, the brain slice was disposed and the perfusion system was thoroughly washed before another brain slice was used for the following experiment.

To assess the effects of fasting on the GABAergic transmission between AgRP neurons and the BNST or between the CeA and the BNST, respectively, fed mice had *ad libitum* access to food. Fasted mice were overnight food deprived for 16 h and sacrificed on the next morning (fasted group). Brain slices from experimental groups were prepared 3 h into the light cycle. To assess the effects of AgRP neuron stimulation, hM3Dq- or mCherry-expressing mice received an i.p. injection of CNO (Clozapine N-oxide -dihydrochloride; 1 mg/kg; Hello Bio, Bristol, UK) 2 h into the light cycle and food was removed from the animals' home cages. Brain slices were prepared 3 h after CNO administration. All recordings were analyzed offline using Clampfit 10.7 (Molecular Devices, Sunnyvale, USA). Currents that were initiated within 20 ms following the onset of the light pulse were considered eIPSCs and cells with reliably reoccurring currents within this time window were considered as connected to the ChR2-expressing population of neurons. For the analysis of eIPSCs, results from 8–10 sweeps were averaged for analysis.

To calculate the coefficient of variation (CV) value or the jitter, the standard deviation of eIPSC amplitude or the latency, respectively, was divided by the mean eIPSC amplitude/latency from the same sweeps.

For analysis of the paired pulse (PP) ratio, 4–5 sweeps were averaged. The PP ratio was calculated as the ratio of the peak amplitude of the second eIPSC divided by the peak amplitude of the first eIPSC (eIPSC2/eIPSC1).

The PP probability represents the ratio of paired light pulses that elicit an eIPSC for both light pulses. This measurement serves as an additional parameter that complements the assessment of the release probability of a synapse. The PP probability is a valuable indicator for synaptic connections, that do not show consistent postsynaptic

responses upon consecutive presynaptic stimulation at higher frequency (e.g., AgRP neuron synapses[38]), and where the PP ratio cannot be calculated.

## Elevated O-maze

The elevated O-maze (EOM) apparatus (constructed in-house) had the following measurements: 50 cm diameter, 5 cm lane width, 15 cm wall height and 40 cm leg height. Before the actual experiments, mice were singly housed and habituated to the experimental room as well as to the handling and the EOM apparatus for at least 10 consecutive days.

**NPY-WT/-KO fed vs. fasted.** On each experimental day, mice of both genotypes were evenly separated into four groups. The mice were either *ad libitum* fed or fasted for 16 h before the beginning of the experiments, and either neutral objects or food pellets were placed on the open arms. The experiments were repeated until each mouse had performed every possible combination of physiological condition (fed/fasted) and stimulus (object/food). In the non-food object condition, two wooden shavings were placed in the center of the open arms. In the food condition, two chow pellets (~3 g) were placed in the center of the open arms.

**NPY-WT/-KO photostimulation of AgRP→BNST.** On each experimental day, mice of both genotypes were evenly separated into two groups with or without optogenetic stimulation (5 min pre-stimulation in home cage without food access followed by photostimulation during the time in the EOM; 20 Hz; 1 s on, 3 s off) of AgRP→BNST projections. In each trial, 2 food pellets (~3 g) were placed in the center on the open arms of the EOM.

Both the non-food object and the food pellet were secured to the apparatus by adhesive putty. Each experimental trial lasted 20 min. and the apparatus was cleaned after each session to prevent the influence of odor. All trials were recorded on video and analyzed using VideoMot 3D Analysis V7.01 software (TSE systems, Berlin, Germany).

For the generation of the EOM heatmaps, we calculated the degree between each point in the original data and a baseline between the center and the end of the second closed arm ("region 4"). Next, we grouped points into 4.5° steps along the 360° circle to obtain 80 subsections. For each subsection, the percentage of measurements of the overall number of measurements was calculated and is shown in the circular EOM heatmap by both color and height, where the height corresponds to the square root of the percentage. The limit of the color scale across the example plots is set to 30% to properly represent the highest included percentages and make the data comparable across heatmaps. We used the original region annotation for each point and annotated each section based on the majority of points belonging to a certain region. In some cases, this led to small differences in the section annotation shown in the inner ring of the heatmap. EOM heatmaps were generated in R (version 4.2.2) using the ggplot2 (version 3.4.2) package.

## In vivo optogenetic stimulation

Before the optogenetic experiments, mice were allowed to recover for at least 1 week post-surgery. Mice with bilateral virus injections were allowed to recover for 3 weeks. They were then put into experimental cages and singly housed. All mice were handled on a daily basis to reduce stress during the subsequent experimental procedures. After ~1 week, they were connected to a fiber-optic patch cord (core diameter 200 μm, numerical AP 0.48; Doric lenses) connected to a rotary joint (Doric lenses), and allowed to adapt to this for another period of 3–4 days. On the experimental day, at the beginning of the light phase, the attached fiber-optic patch cord was replaced by a new one. A laser power of 20 mW was used in the ARC and 5 mW for bilateral stimulation in the BNST, rendering an irradiance of ~3–7 mW/mm² and ~1.9–6 mW/mm² in the targeted regions, as calculated with the online

tool at https://web.stanford.edu/group/dlab/cgi-bin/graph/chart.php; hence, above the threshold for activation of ChR2 (~1 mW/mm²). Stimulation protocol and stimulation frequency was adapted between experiments.

## Food intake

All animals were singly housed and handled for at least seven consecutive days before the assay to acclimate mice to the experimental procedure. Feeding studies were performed in home cages with ad libitum food access to chow. Before the experiment, mice were provided with fresh cages to avoid leftover food spilling in the bedding.

For refeeding experiments, mice were provided with fresh cages 1 h before onset of the dark cycle on the day before the experiment and no food was provided. After 16 h fasting, food was placed back into the home cage and mice had *ad libitum* food access.

For measuring food intake upon chemogenetic activation of AgRP neurons, clozapine N-oxide (CNO) was diluted in saline and administered at 1 mg/kg of body weight 3 h after onset of the light cycle. Mice had either direct ad libitum access to food or 3 h after injection.

For measuring food intake during optogenetic stimulation of AgRP^ChR2 or CeA^ChR2 terminals in the BNST, a laser power of 20 mW was used, rendering an irradiance of ~3–7 mW/mm² in the targeted region. AgRP^ChR2 terminals were either continuously stimulated at 5 Hz or at 20 Hz (1 s ON, 3 s OFF), CeA^ChR2 terminals were continuously stimulated at 2.5 Hz, 5 Hz, 7.5 Hz, or 10 Hz.

For dark-cycle experiments, measurement of food intake was started 1 h before the end of the light cycle. Food intake was monitored for up to 5 h.

## Organ tissue preparation

For organ collection, all mice were deeply anesthetized with ketamine and xylazine, and euthanized with transcardial perfusion of phosphate buffered saline (PBS), followed by 4% paraformaldehyde (PFA) in PBS (PFA-PBS). Brains were dissected, post-fixed at 4 °C in PFA-PBS for 16–22 h and then transferred to 20% sucrose in PBS. Brains were cut in 20–30 μm sections for immunostaining using a microtome. To cover the rostro-caudal axis of the brain region of interest, every fourth section was further processed (see below). The residual sections were collected in cryoprotectant and stored at −20 °C.

For post hoc analysis of virus expression and optic fiber placement, brain slices of electrophysiological and in vivo experiments were collected, and fixed in PFA-PBS before they were washed and stored at 4 °C in PBS.

## Immunohistochemistry

Sections were blocked with 2% normal donkey serum in 0,4% Triton X-100 in PBS (NDS-PBST) for 1 h at room temperature (RT) and incubated with primary antibodies (goat anti-AgRP, Neuromics, Minneapolis, USA) diluted in NDS-PBST overnight at RT. Sections were washed with PBST and then incubated with secondary antibodies diluted in PBS for 1 h at RT. After several washing steps with PBS, sections were mounted and counterstained with DAPI containing mounting medium (VECTASHIELD® Antifade Mounting Medium with DAPI, Cat# H-1200, Vector Laboratories).

## In situ hybridization

RNAscope Multiplex Fluorescent Reagent Kit v2 (Advanced Cell Diagnostic, Cat# 323100) was used following the manufactures' instructions. Sections were dried at 60 °C overnight, pre-treated with hydrogen peroxide (Cat# 322381), and boiled in Target retrieval (Cat# 322000). After dehydrating in pure ethanol, sections were surrounded by a hydrophobic barrier (ImmEdge hydrophobic barrier pen, Vector Lab, H-4000) and incubated in Protease Plus (Cat# 322331; 15 min. at 40 °C) followed by the target probe (tdTomato-C2, Cat# 317041-C2) in

**Article**

a HybEZoven. Signal amplification was reached using amplifiers AMP1-3 and label probe (Cy3; Perkin-Elmer, Cat#NEL760001KT). Sections were mounted using DAPI containing mounting medium (VECTA-SHIELD, Cat# H-1200, Vector Laboratories).

### Imaging

Brain sections were imaged by a Keyence BZ-9000E fluorescent microscope (Keyence) with 4 x or 20 x magnification, Zeiss ImagerM2 fluorescent microscope with 4 x magnification, or a Leica Stellaris 8 Falcon Confocal Microscope (Leica Microsystems) with 20 x and 63 x magnification. Images were further processed and fluorescence intensity was quantified using ImageJ software (Schneider, Rasband, & Eliceiri, 2012) and is based on 3–6 BNST-containing brain sections per animal.

### Statistical analysis

Statistical analyses were performed using Prism 9.3.1 (GraphPad) software. Statistical tests applied are found in the figure legends. No statistical method was used to predetermine sample size. Sample sizes are reported in the figure legends. The Kolmogorov-Smirnov test was used to test for normality of data distribution. Blinding methods were not used. All data presented met the assumptions of the statistical test employed. Bar graphs show average connectivity rates or mean +/- standard error of mean (SEM). Violin plots show median (line), quartiles (dashed lines) and the range of the distribution. Pie charts show relative distributions. Statistical significance is represented by $*p < 0.05$, $**p < 0.01$, $***p < 0.001$, and $****p < 0.0001$.

### Reporting summary

Further information on research design is available in the Nature Portfolio Reporting Summary linked to this article.

## Data availability

Source data are provided with this paper as a Source Data file. Any additional information required to reanalyze the data reported in this paper is available from the lead contact upon request. Source data are provided with this paper.

## Code availability

No original code was generated in this study.

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

## Acknowledgements

We thank all members of the Fenselau lab and Brüning lab for helpful discussions, as well as Ursula Lichtenberg, Karina Schöfisch, Hella Brönneke, and Ramona Braun for their excellent administrative help. H.F. received funding within the Excellence Initiative by German Federal and State Governments (CECAD). J.C.B and H.F. received funding within the European Research Council grant agreement (# 742106 and # 851778, respectively). S.D. received funding from the Deutsche Forschungsgemeinschaft (DFG, German Research Foundation; 214362475/GRK1873/2). We thank Prof. Dr. Annette Beck-Sickinger for providing the NPY2R agonist Ahx[5-24]-NPY.

## Author contributions

S.D., J.C.B., and H.F. conceived the project. S.D., N.V.W., M.L.D., L. W., L.S., F.T.W. planned and performed the experiments, and analyzed the data. S.D. and H.F. wrote the manuscript. All authors revised and approved the manuscript.

## Funding

## Competing interests

The authors declare no competing interests.
