## [Peer Review File · Nature Communications]

NPY-mediated synaptic plasticity in the extended amygdala prioritizes feeding during starvationREVIEWER COMMENTS

Reviewer #1 (Remarks to the Author):

In this manuscript, Dodt et al explore NPY-sensitive BNST circuits in fasted and fed states in mice. They present lines of data indicating changes in connectivity in AGRP- and CeA-BNST synapses in these states, with data suggesting these changes are regulated by the NPY signaling system. They propose an NPY mediated plasticity in the BNST. The results add to a growing literature on the roles of the NPY and BNST signaling. The investigators use a wide array of approaches to contribute to this literature. The results will be of interest to the feeding behavior/neural circuit field, but I do have a number of significant concerns regarding aspects of the work that need to be considered.

- 1) A significant concern is the region targeted in this study. From the images shown throughout this manuscript, the regions sampled and targeted appear likely to include a significant amount of accumbens relative to BNST. Significantly improved visual validation of BNST targeting is needed. Related to this, more discussion needs to be provided regarding the subregions of the BNST sampled in the electrophysiological recordings. The extremely low level of basal connectivity, as the authors state, is very surprising given the prior demonstrations in the literature of massive AGRP projections to the BNST, as well as functional effects of stimulating this pathway, leading to questions regarding anatomical sampling choices.
- 2) The quality of the histochemistry images and the magnification is insufficient to visually assess relative colocalization of signals. Higher resolution images would likely be sufficient to rectify this. In some cases, the authors seem to make conclusions regarding synaptic contacts based on fiber fluorescence in BNST. This is an over-reach and should be removed except in cases where functional coupling and/or more specific markers like synaptophysin are utilized.
- 3) The sex of mice utilized is not consistently mentioned, and where it is mentioned, no sense is given as to whether they were equally included and/or whether any obvious differences in the data were observed. To be clear, I am not requesting that the authors power for a sex differences study, just that they more rigorously report the composition of the samples in the reported work. This is particularly important since the literature suggests sex-dependent differences in melanocortin receptor expression.
- 4) The comparison of AGRP-BNST to CeA-BNST GABAergic function is interesting, but care needs to be taken in extrapolating conclusions. While the CeA input appears stronger in these studies, the CeA is known to be one of the largest inputs to the BNST. En masse optogenetic activation of all CeA inputs vs all AGRP inputs is unlikely to equate to what physiological recruitment of smaller ensembles of these fibers would do, where the differences shown here due to higher number of raw axonal inputs is likely to dominate. Another weakness in this comparison is that a somewhat unitary population from the hypothalamus is queried (AGRP), while likely multiple CeA cell types are lumped together. A further weakness is that one population is assessed via ChR2 expressed through genetic crosses, where the other is generated by viral delivery. Finally, as mentioned above, the low level of AGRP connectivity is confusing given the massive innervation from AGRP cells. It would be useful to demonstrate that with the vectors and approaches used here, higher GABAergic connectivity on cells in other AGRP target fields are observed as previously reported in the literature. Another useful approach might be to image vGAT+ puncta within AGRP tracer labelled fibers in BNST.
- 5) The fed/fasting physiology in figure 3 is somewhat difficult to assess. A fasting induced increase in connectivity but decrease in GABAergic transmission is a very unusual potential phenotype. I find the amplitude data convincing, but am worried about the connectivity difference, given that the connectivity difference refers to 8/157 (5%) relative to 20/167 (12%). The very low efficiency here leads to concerns that these subtle differences in connectivity may be driven by sampling differences.
- 6) The NPY histochemistry difference in figure 4 needs to be cross validated with an in situ approach and/or some other measure, as the current data alone are not sufficiently convincing. Moreover, its not apparent necessarily where the source of this NPY is. NPY quantification in ROIs with AGRP labelled fibers would help.
- 7) The presentation of data in figure 5b is difficult to understand. It seems like a heatmap

approach would be more intuitive. Further, more, or clearer, information needs to be provided as it's not clear what the order of experiments was in this figure, and the extent to which within subject analyses were performed, and if so whether a latin square design was used.

8) Previous studies have demonstrated that NPY decreases IPSCs in BNST (for example Kash and Winder, *Neuropharmacology* 2006). The present data extend these with synapse specificity, but the original data should be cited.

9) In my opinion, the authors too freely use the term "plasticity", when "modulation" seems more appropriate. For plasticity to be appropriate, I'd want to see longer term persistence of the electrophysiological changes, and corroborating anatomical data.

Reviewer #2 (Remarks to the Author):

This MS reports on studies of the question how animals balance hunger and fear responses in the face of starvation vs predation. Focusing on activity in arcuate nucleus AgRP/NPY/GABA (AgRP- "hunger" neurons) neurons and central amygdala (CeA - "fear" neurons) that make GABAergic inhibitory synapses with neurons in the Bed Nucleus of the Stria Terminalis (BNST - which integrates the different drives and mediates both eating and avoidance), they used transgenic, optogenetic, electrophysiological and behavioral approaches to examine the effect of starvation on the synaptic responses evoked in BNST neurons from the two input nuclei. Fasting increased the number of BNST neurons that responded to AgRP stimulation, while input from the CeA is reduced in amplitude. Stimulation of AgRP neurons increases feeding, while activation of CeA suppresses appetite.

Behaviorally, prolonged activity in AgRP neurons is sufficient to cause the hunger-associated reduction in response to CeA stimulation. This reduction is eliminated in NPY-knockout mice, implicating NPY in mediating the changes. NPY-deficient mice also show no reduction in anxiety-related behavior. They conclude that this essential risk-benefit analysis requires the action of NPY in the BNST to mediate plastic changes that facilitate food-seeking in the presence of a threat. This is a really neat paper, which I enjoyed reading and which addresses an important question in neurobiology. The experiments are well-designed and -executed, and the manuscript is generally well- and clearly-written. However, there are some small grammatical and idiomatic suggestions I have made in the accompanying marked-up MS.

There are several questions I would like the authors to please clarify:

1) I understand that the transgenic approach they chose here is well-established and convenient in their hands (e.g. NPY knockout mouse, tg AgRP-cre X tg flox Chr2-Eyfp, etc.), vs. doing retrograde and intersectional viral targeting of specific pathways, etc. Were they at all concerned about (or how did they control for) activation of collaterals during optogenetic stimulation in vivo, not so much with the AgRP cells but with the CeA neurons? This would include the possibility of backpropagation to excite these other branches with the in vivo optogenetic stimulation.

2) Their choice of NPY 13-36 is less than ideal, as it activates both Y5Rs and Y2Rs. Much cleaner for Y2R is the [ahx5-24]NPY agonist from the Beck-Sickinger lab in Leipzig. Alternatively, they could block the effect of NPY by pre-incubation with any of the high-affinity Y2R antagonists available. They should also test the Y5R antagonist CGP71683, which is available from Tocris

3) Re: final sentence of the Results - Did they record the spontaneous synaptic activity of the BNST neurons in ex-vivo slices, as NPY might activate Y2Rs on presynaptic GABAergic or glutamatergic terminals, and indirectly affect BNST neurons.

4) Methods- With the optogenetic stimulation in vitro, did they check the stimulus-response relationship of the synaptic inputs by altering the luminous output of the laser, or changing the duration of the optical stimulus to determine where they were relative to the maximum output of a given synapse? The responses from CeA stimulation are very large (nA). It is clear that strongly stimulating synapses will tend to overcome any acute modulatory actions of presynaptic inhibitory receptors, such as NPYRs.

5) Please explain in the methods how they measured paired-pulse probability, a measure I am not familiar with.

6) There are numerous instances in figures (and extended data) where the individual symbols are vanishingly small, please increase the size of these throughout. Also traces (Fig 3b, e, Fig 4d, g,

Fig 6 c, d, e) should be thickened to the point where they can be more clearly seen without zooming in a lot. I have annotated a number of these in the MS.

Minor concerns-

- 1) In the methods (p. 23, Elevated O-maze), the authors refer to introducing a home cage scent by the transfer of wooden bedding "sticks" – do they mean wood shavings? A bed of sticks sounds distinctly uncomfortable!
- 2) Individual comments regarding figures are marked on the figures themselves - at least one typo appears there as well (Fig 4A Y-Axis).

Reviewer #3 (Remarks to the Author):

In this study, the authors investigate the actions of hunger and NPY, an orexigenic peptide, within the BNST, a region involved in both anxiety and feeding. The study focuses on central amygdala inputs to BNST, a projection which has been previously implicated in anxiety and feeding. Using behavioral assays, genetic manipulations, and ex vivo slice electrophysiology approaches, they characterize the major source of NPY in BNST, AgRP inputs, and its effect on CeA inputs. The authors find weak synaptic connectivity of AgRP inputs despite an apparently strong behavioral consequences of this projection. Interestingly, fasting induced increases in NPY in BNST, and both fasting and NPY from AgRP neurons could decrease the strength of CeA inputs to BNST, an effect that was eliminated in whole body genetic KO of NPY. Further experiments then implicate changes at CeA to BNST synapses in mediating the anxiolytic effects of hunger, but direct evidence that NPY from AgRP neurons is required for this feeding effect is missing. This original study is timely as there is considerable interest in the field for the important actions of NPY on synaptic properties. The approaches and methods are overall sound, though some major arguments in the study could be strengthened as indicated in comments below.

Major comments:

- 1) The in vivo optogenetics stimulation frequency plays an important role in the arguments in this study, but there is need for more clarity on some details. Can the authors include reasoning for the differences in stimulation paradigm used between ARC (20 mW, 20 Hz, 1s ON 3s OFF) vs CeA (5 mW, continuous rather than ON-OFF stimulation) experiments and how the choice of CeA stimulation might affect results? For example, would similar findings be expected for CeA stimulation using a 1s ON, 3s OFF protocol?
- 2) The finding that AgRP inputs to BNST are onto non-specific BNST neurons would be strengthened by more anatomical information about cell locations within the BNST. Specifically, can the authors show the location of each neuron recorded within the extended BNST overlaid on atlas images so that one can determine if connectivity rate is biased to regions of the BNST (for example, along the A-P or D-V axis of BNST). Furthermore, the histological images used in the figures are small and cropped making it difficult to guess where in BNST these studies are performed.
- 3) In Figure 3H, it's unclear what the asterisks are indicating. Please clarify in figure legends and/or edit the plot to clearly indicate what comparisons are being made that the asterisks refer to.
- 4) In Figure 5, the authors show that NPY KO eliminates the anxiolytic effect of hunger in the EOM assay, but it does not directly show that the anxiolytic effects of AgRP stimulation is mediated by NPY. In Figure 6, the authors nicely demonstrate that the synaptic effects of AgRP evoked NPY are absent in NPY KO mice, but this does not directly tie back to the in vivo feeding results. The strongest evidence of this would be showing that AgRPBNST photostimulation in the EOM assay recapitulates how hunger increases feeding in the open areas and that this effect is absent in AgRP photostimulation in NPY-KO mice.
- 5) The authors state that NPY-KO mice are not significantly different in the first 30 minutes of a re-feeding assay, but they do not acknowledge that they are significant at the 60-minute mark (Ext Fig 5). Furthermore, even in Extend Fig 5d, there does seem to be a reduction at 30 minutes but appears to be underpowered for statistical significance. I think this is important to address and discuss as the authors are using this as evidence of context-specific actions of NPY. To me it seems there are some global effects on feeding based on the data.

6) The statement that NPY does not affect neuronal activity of BNST neurons (Figure 6G) is suggestive and indirect. If these neurons are spontaneously active under cell-attached recording conditions, it would be better to record spontaneous firing under these less perturb conditions and show if NPY decreases firing rate of BNST neurons.

7) This discussion would benefit from a schematic/diagram of the model for how NPY influences the BNST circuits to impact anxiety and feeding would help. In particular, do the authors hypothesize that the increased synapse number of AgRP neurons onto BNST neurons actually matters for the anxiolytic effects observed or only an indirect consequence and that general extracellular release of NPY and action onto CeA presynaptic terminals is the actual mechanism? Can the authors discuss if there is expectation that CeA inputs would be near AgRP inputs or is it possible that diffusion of the peptide might be action on CeA inputs on totally different BNST neurons than the ones the AgRP neurons contact.

Minor comments:

1) In the introduction, in the sentence "AgRP neuron activation decreases anxiety-related behavior and increases risk behavior to maximize food acquisition", could also cite Jikomes et al., 2016 DOI: 10.1016/j.cub.2016.07.019

2) Typo in methods section: "ad libidum" should be "ad libitum"

3) Can the authors clarify in the methods why 20 mW light was used for ARC experiments and whether the statement "rendering an irradiance of ~3-7 mW/mm²" was only for the BNST. This is unclear in two locations in the methods text whether the authors are saying 20 mW ARC stim was only leading to 3-7 mW of light reaching the ARC. Can authors calculate and state what irradiance for ARC would be given 20 mW was used?

4) Light On controls (i.e. mice receiving photostimulation but not expressing ChR2) would have been nice controls for CeA \rightarrow BNST projections.

5) Figure 2: please indicate in legends the number of slices in addition to the number of mice that the recordings came from. In other words, for the N=13 AgRP mice in 2C, how many slices were recorded from per mouse.

6) A limitations of study section in the discussion would be nice and include whether CeA neurons collateralize and if this might contribute to the results. In addition, the choice of stimulation protocol as mentioned in major comment 1.

Response to Reviewers

„NPY-mediated synaptic plasticity in the extended amygdala prioritizes feeding during starvation”
(NCOMMS-23-40493-T)

We wish to thank the Reviewers for their constructive and insightful comments on our manuscript. As described in detail in this document, we have performed an extensive series of additional, critical experiments to directly address the Reviewers' concerns. We believe that incorporating our additional findings, and the recommended modifications to the text, have significantly improved our manuscript.

Below, we detail how each of the Reviewers' concerns was addressed. Each concern is highlighted in bold, immediately followed by the corresponding response in italic. Given the extent of our additional findings, which led to editing many parts of text and figures, we describe where information addressing the Reviewers' concerns can be found.

Reviewers' comments:

Reviewer #1:

In this manuscript, Dodt et al explore NPY-sensitive BNST circuits in fasted and fed states in mice. They present lines of data indicating changes in connectivity in AGRP- and CeA-BNST synapses in these states, with data suggesting these changes are regulated by the NPY signaling system. They propose an NPY mediated plasticity in the BNST. The results add to a growing literature on the roles of the NPY and BNST signaling. The investigators use a wide array of approaches to contribute to this literature. The results will be of interest to the feeding behavior/neural circuit field, but I do have a number of significant concerns regarding aspects of the work that need to be considered.

We thank the Reviewer for the positive assessment of our manuscript and the constructive comments.

1) A significant concern is the region targeted in this study. From the images shown throughout this manuscript, the regions sampled and targeted appear likely to include a significant amount of accumbens relative to BNST. Significantly improved visual validation of BNST targeting is needed. Related to this, more discussion needs to be provided regarding the subregions of the BNST sampled in the electrophysiological recordings. The

extremely low level of basal connectivity, as the authors state, is very surprising given the prior demonstrations in the literature of massive AGRP projections to the BNST, as well as functional effects of stimulating this pathway, leading to questions regarding anatomical sampling choices.

We completely agree that our report would benefit from an improved visual presentation of the targeted brain regions. Therefore, we now include representative images of the BNST as well as the nucleus accumbens. Our additional images show that projections from both AgRP neurons and CeA neurons densely innervate the BNST, including its anterior subnucleus, but do not target the nucleus accumbens (shown in Extended data Fig. 1a and b).

We would like to highlight that we found these projection patterns also in all electrophysiological recordings that we performed by assessing expression of the ChR2-fused fluorophores. In addition, we always confirmed that we recorded from neurons in the anterior BNST by orienting on specific landmarks, such as the 3rd ventricle, the shape as well as the size of the anterior commissure, and the location of the lateral ventricle. To illustrate this, we now include representative schematic images illustrating the location of individual neurons from which we performed electrophysiological recordings (shown in Extended data Fig. 3a). As suggested by the Reviewer, we now also include a statement on the functional relevance of this subregion of the BNST that we investigated in the discussion section (page 15, line 14 - 16).

2) The quality of the histochemistry images and the magnification is insufficient to visually assess relative colocalization of signals. Higher resolution images would likely be sufficient to rectify this. In some cases, the authors seem to make conclusions regarding synaptic contacts based on fiber fluorescence in BNST. This is an over-reach and should be removed except in cases where functional coupling and/or more specific markers like synaptophysin are utilized.

To address this important point, we obtained additional new images with higher resolution microscopy, which we include in our revised version of the manuscript to better visually describe colocalization of fluorescent signals (Fig. 1a, b; Fig. 2d; extended data Fig. 2f; Fig. 4a, b; extended data Fig. 4a, b). In addition, we changed the wording in the manuscript with regards to the synaptic contacts in the BNST (page 5, line 7-9).

3) The sex of mice utilized is not consistently mentioned, and where it is mentioned, no sense is given as to whether they were equally included and/or whether any obvious differences in the data were observed. To be clear, I am not requesting that the authors power for a sex differences study, just that they more rigorously report the composition of

the samples in the reported work. This is particularly important since the literature suggests sex-dependent differences in melanocortin receptor expression.

We thank the Reviewer for bringing up this important point. We now include a detailed statement on the sex composition of the mice that were used for the individual experiments in the methods section of the manuscript (page 20, line 10 - 13).

4) The comparison of AGRP-BNST to CeA-BNST GABAergic function is interesting, but care needs to be taken in extrapolating conclusions. While the CeA input appears stronger in these studies, the CeA is known to be one of the largest inputs to the BNST. En masse optogenetic activation of all CeA inputs vs all AGRP inputs is unlikely to equate to what physiological recruitment of smaller ensembles of these fibers would do, where the differences shown here due to higher number of raw axonal inputs is likely to dominate. Another weakness in this comparison is that a somewhat unitary population from the hypothalamus is queried (AGRP), while likely multiple CeA cell types are lumped together. A further weakness is that one population is assessed via ChR2 expressed through genetic crosses, where the other is generated by viral delivery. Finally, as mentioned above, the low level of AGRP connectivity is confusing given the massive innervation from AGRP cells. It would be useful to demonstrate that with the vectors and approaches used here, higher GABAergic connectivity on cells in other AGRP target fields are observed as previously reported in the literature. Another useful approach might be to image vGAT+ puncta within AGRP tracer labelled fibers in BNST.

The Reviewer raises an important point regarding the activation of a defined population of AgRP neurons and a heterogeneous population of GABAergic neurons of the CeA. We now include an additional limitation section to our discussion (page 17, line 1 – 11).

To investigate whether the low connectivity rate of the AgRP → BNST circuit is due to the potentially lower expression level of ChR2 in our transgenic mouse model, we performed additional experiments with AAV-mediated expression of ChR2 in AgRP neurons. Consistent with our findings in transgenic mice, the connectivity rate was comparably low in mice expressing ChR2 virally (Extended Data Fig. 2d).

In addition, as suggested by the Reviewer, we also performed electrophysiological recordings from PVH neurons in transgenic mice expressing ChR2 in AgRP neurons. As previously reported, we found that approximately 35% of the neurons in the PVH received monosynaptic GABAergic input from AgRP neurons (Atasoy et al., 2012; Ruud et al., 2020). We now include these important additional findings in our new Extended Data Fig. 2c.

5) The fed/fasting physiology in figure 3 is somewhat difficult to assess. A fasting induced increase in connectivity but decrease in GABAergic transmission is a very unusual potential phenotype. I find the amplitude data convincing, but am worried about the connectivity difference, given that the connectivity difference refers to 8/157 (5%) relative to 20/167 (12%). The very low efficiency here leads to concerns that these subtle differences in connectivity may be driven by sampling differences.

We thank the Reviewer for this comment. To validate that the increase in connectivity rates that we observed was not due to sampling differences, we deliberately performed electrophysiological recordings from a high number of neurons in both fed and fasted mice. In addition, we found that NPY-WT mice, but not NPY-KO mice, exhibited an almost identical increase in connectivity rate when AgRP neurons were chemogenetically stimulated (Fig. 6c). We would like to note that all of our recordings were performed from the same BNST region as we now also depict in our new schematic shown in the Extended Data Fig. 3a. Together, these data strongly support our conclusion that activation of AgRP neurons results in an increased connectivity of the AgRP → BNST circuit in an NPY-dependent manner.

6) The NPY histochemistry difference in figure 4 needs to be cross validated with an in situ approach and/or some other measure, as the current data alone are not sufficiently convincing. Moreover, it's not apparent necessarily where the source of this NPY is. NPY quantification in ROIs with AGRP labelled fibers would help.

We thank the Reviewer for raising this point. We would like to point out that numerous previous studies have shown that energy deprivation markedly increases NPY expression in AgRP neurons at the level of their cell bodies in the ARC (Hahn et al, 1998; Luquet et al., 2005; Schwartz et al., 1992), as well as in their projections to the PVH (Cabral et al., 2020). In addition, our data (Fig. 4b and Extended Data Fig. 4a) and previous findings (Kash et al., 2015) demonstrate that the vast majority of NPY in the BNST derives from AgRP neurons. To validate this important aspect again, we performed additional histological experiments. We stained brains from AgRP^{ChR2} mice for NPY and EYFP, which allowed us to determine the co-expression of ChR2 from AgRP neurons and NPY independent of AgRP, which may be altered under energy-deprived conditions. These additional experiments confirmed that the vast majority of NPY-positive fibers are AgRP neuron

terminals (Fig. 4b and Extended Data Fig. 4a). Nevertheless, we discuss the possibility that increased NPY expression levels might also arise from other neuronal populations, such as NPYergic BNST interneurons (page 15, line 21 – page 16, line 6).

7) The presentation of data in figure 5b is difficult to understand. It seems like a heatmap approach would be more intuitive. Furthermore, or clearer, information needs to be provided as it's not clear what the order of experiments was in this figure, and the extent to which within subject analyses were performed, and if so whether a latin square design was used.

We completely agree with the Reviewer and have accordingly adjusted Figure 5B. In addition, we now included further information to Fig. 5b to provide additional clarification of the presented data. Further, we now provide detailed information on the order of experiments is in an additional paragraph in the methods section (page 25, line 4 - 11; page 25, line 13 – 18).

8) Previous studies have demonstrated that NPY decreases IPSCs in BNST (for example Kash and Winder, Neuropharmacology 2006). The present data extend these with synapse specificity, but the original data should be cited.

We thank the Reviewer for this remark. We added this relevant citation to our results (page 14, line 12) as well as discussion sections (page 17, line 16).

9) In my opinion, the authors too freely use the term "plasticity", when "modulation" seems more appropriate. For plasticity to be appropriate, I'd want to see longer term persistence of the electrophysiological changes, and corroborating anatomical data.

We agree with the Reviewer that both terms, plasticity and modulation, can in principle be used. However, as our study primarily focused on transmission of GABAergic synapses, we think that plasticity is more suitable as modulation is often used in the context of neuropeptidergic transmission. We have accordingly adjusted the text.

Reviewer #2:

This MS reports on studies of the question how animals balance hunger and fear responses in the face of starvation vs predation. Focusing on activity in arcuate nucleus AgRP/NPY/GABA (AgRP- "hunger" neurons) neurons and central amygdala (CeA – "fear" neurons) that make GABAergic inhibitory synapses with neurons in the Bed Nucleus of the Stria Terminalis (BNST – which integrates the different drives and mediates both eating and avoidance), they used transgenic, optogenetic, electrophysiological and behavioral approaches to examine the effect of starvation on the synaptic responses evoked in BNST neurons from the two input nuclei. Fasting increased the number of BNST neurons that responded to AgRP stimulation, while input from the CeA is reduced in amplitude. Stimulation of AgRP neurons increases feeding, while activation of CeA suppresses appetite.

Behaviorally, prolonged activity in AgRP neurons is sufficient to cause the hunger-associated reduction in response to CeA stimulation. This reduction is eliminated in NPY-knockout mice, implicating NPY in mediating the changes. NPY-deficient mice also show no reduction in anxiety-related behavior. They conclude that this essential risk-benefit analysis requires the action of NPY in the BNST to mediate plastic changes that facilitate food-seeking in the presence of a threat.

This is a really neat paper, which I enjoyed reading and which addresses an important question in neurobiology. The experiments are well-designed and –executed, and the manuscript is generally well- and clearly-written. However, there are some small grammatical and idiomatic suggestions I have made in the accompanying marked-up MS. There are several questions I would like the authors to please clarify:

We would like to thank the Reviewer for the very positive assessment of our manuscript. We greatly value the careful reading and the detailed comments.

1) I understand that the transgenic approach they chose here is well-established and convenient in their hands (e.g. NPY knockout mouse, tg AgRP-cre X tg flox Chr2-Eyfp, etc.), vs. doing retrograde and intersectional viral targeting of specific pathways, etc. Were they at all concerned about (or how did they control for) activation of collaterals during optogenetic stimulation in vivo, not so much with the AgRP cells but with the CeA neurons? This would include the possibility of backpropagation to excite these other branches with the in vivo optogenetic stimulation.

The Reviewer raises an important concern which we now acknowledge in the limitations section of our discussion (page 18, line 11-16).

2) Their choice of NPY 13-36 is less than ideal, as it activates both Y5Rs and Y2Rs. Much cleaner for Y2R is the [ahx5-24]NPY agonist from the Beck-Sickinger lab in Leipzig. Alternatively, they could block the effect of NPY by pre-incubation with any of the high-affinity Y2R antagonists available. They should also test the Y5R antagonist CGP71683, which is available from Tocris.

We wish to thank the Reviewer for this comment. To validate the involvement of the NPY2R, we used the suggested, highly selective agonist NPY2R receptor agonist Ahx[5-24]NPY. We found that bath application of this compound resulted in an identical inhibition of GABAergic transmission between the CeA and the BNST. These findings further support our initial conclusion of a NPY2R-mediated attenuation of GABAergic transmission. We have included these additional important findings in our new Fig. 6f.

3) Re: final sentence of the Results - Did they record the spontaneous synaptic activity of the BNST neurons in ex-vivo slices, as NPY might activate Y2Rs on presynaptic GABAergic or glutamatergic terminals, and indirectly affect BNST neurons.

We thank the Reviewer for this comment. To further investigate the role of NPY signaling in the regulation of spontaneous synaptic activity in the BNST, we have performed additional pharmacological experiments on BNST neurons in ex vivo slices and concentrated our experiments on the investigation on the GABAergic transmission in the BNST. We found that bath application of both NPY and NYP13-36 caused a reduction in sIPSC frequencies whereas there were not significant changes in sIPSC amplitudes. We now include these additional findings in Extended Data Fig. 6g, h.

4) Methods- With the optogenetic stimulation in vitro, did they check the stimulus-response relationship of the synaptic inputs by altering the luminous output of the laser, or changing the duration of the optical stimulus to determine where they were relative to the maximum output of a given synapse? The responses from CeA stimulation are very large (nA). It is clear that strongly stimulating synapses will tend to overcome any acute modulatory actions of presynaptic inhibitory receptors, such as NPYRs.

We thank the Reviewer for pointing out this important concern. In all of our experiments, we used the maximum output of the LEDs and did not change the duration of the light pulses. Thus, it could be that this resulted in strong synapse stimulation. However, in all of our experimental settings,

including fasting, chemogenetic AgRP neuron stimulation, and addition of NPY receptor ligands, we observed strong reductions in transmission across the CeA→BNST synapses. Thus, we are certain that, despite the maximum LED intensity, neuromodulatory effects were not profoundly masked.

5) Please explain in the methods how they measured paired-pulse probability, a measure I am not familiar with.

A more detailed explanation for the assessment of the paired pulse probability is now included in the methods section (page 24, line 15 - 20).

6) There are numerous instances in figures (and extended data) where the individual symbols are vanishingly small, please increase the size of these throughout. Also traces (Fig 3b, e, Fig 4d, g, Fig 6 c, d, e) should be thickened to the point where they can be more clearly seen without zooming in a lot. I have annotated a number of these in the MS.

We thank the Reviewer for bringing this up. Accordingly, we adjusted size of symbols and traces in all figures.

Minor concerns:

1) In the methods (p. 23, Elevated O-maze), the authors refer to introducing a homecage scent by the transfer of wooden bedding "sticks" – do they mean wood shavings? A bed of sticks sounds distinctly uncomfortable!

We thank the Reviewer for highlighting the confusion. We revised the corresponding sections.

2) Individual comments regarding figures are marked on the figures themselves - at least one typo appears there as well (Fig 4A Y-Axis).

We greatly appreciate this valuable input. The individual comments are now incorporated.

Reviewer #3:

In this study, the authors investigate the actions of hunger and NPY, an orexigenic peptide, within the BNST, a region involved in both anxiety and feeding. The study focuses on central amygdala inputs to BNST, a projection which has been previously implicated in anxiety and feeding. Using behavioral assays, genetic manipulations, and ex vivo slice electrophysiology approaches, they characterize the major source of NPY in BNST, AgRP inputs, and its effect on CeA inputs. The authors find weak synaptic connectivity of AgRP inputs despite apparently strong behavioral consequences of this projection. Interestingly, fasting induced increases in NPY in BNST, and both fasting and NPY from AgRP neurons could decrease the strength of CeA inputs to BNST, an effect that was eliminated in whole body genetic KO of NPY. Further experiments then implicate changes at CeA to BNST synapses in mediating the anxiolytic effects of hunger, but direct evidence that NPY from AgRP neurons is required for this feeding effect is missing. This original study is timely as there is considerable interest in the field for the important actions of NPY on synaptic properties. The approaches and methods are overall sound, though some major arguments in the study could be strengthened as indicated in comments below. Major comments:

We would like to thank the Reviewer for the very positive assessment of our manuscript and the comments.

1) The in vivo optogenetics stimulation frequency plays an important role in the arguments in this study, but there is need for more clarity on some details. Can the authors include reasoning for the differences in stimulation paradigm used between ARC (20 mW, 20 Hz, 1s ON 3s OFF) vs CeA (5 mW, continuous rather than ON-OFF stimulation) experiments and how the choice of CeA stimulation might affect results? For example, would similar findings be expected for CeA stimulation using a 1s ON, 3s OFF protocol?

We completely agree with the Reviewer that additional information on this important aspect of our studies is warranted. For the stimulation of AgRP neurons we chose a well-established protocol of 20 Hz (1 sec. ON/3 sec. OFF) because previous studies from our and other laboratories have shown that it evokes food intake increase as well as insulin resistance (Aponte et al., 2011; Steculorum et al., 2016; Ruud et al., 2020). Importantly, this optogenetic stimulation protocol has also been shown to induce these adaptations in an NPY-dependent manner (Chen et al., 2019; Ruud et al., 2020), a critical prerequisite for our studies. For our investigation of GABAergic transmission of the CeA → BNST circuit, we employed a continuous optogenetic stimulation protocol as this was found to evoke behavioral changes (Cai et al., 2014; Botta et al., 2015). In

our revised manuscript, we now include additional information on this important aspect (page 5, line 16 - 19).

In addition, to reexamine whether a continuous, low frequency stimulation (5 Hz) would also be sufficient to increase food intake when ChR2 is expressed in AgRP neurons, we performed additional experiments in a new cohort of mice. Our new data, which are shown in Extended Data Fig. 1d, demonstrate that this optogenetic stimulation protocol, which lowers food intake in CeA^{ChR2} mice, indeed stimulates food intake in AgRP^{ChR2} mice. This further supports our conclusion that GABAergic AgRP→BNST and CeA→BNST inputs have mechanistically and functionally distinct features through which they control BNST neurons.

2) The finding that AgRP inputs to BNST are onto non-specific BNST neurons would be strengthened by more anatomical information about cell locations within the BNST. Specifically, can the authors show the location of each neuron recorded within the extended BNST overlaid on atlas images so that one can determine if connectivity rate is biased to regions of the BNST (for example, along the A-P or D-V axis of BNST). Furthermore, the histological images used in the figures are small and cropped making it difficult to guess where in BNST these studies are performed.

We thank the Reviewer for pointing out these important concerns. To determine the approximate location of the BNST that we investigated, we have performed additional analyses of our original recordings for which we had documented the neurons' location. We now include representative figures in the Extended Data Fig. 3a which illustrate our findings.

In addition, we obtained new images from our histological experiments with higher resolution microscopy. We include these additional images in our revised version of the manuscript to better describe the anatomical location of the BNST regions that we investigated (Fig. 1a, b; Extended Data Fig. 1a, b; Extended Data Fig. 2f; Fig. 4a, b; Extended Data Fig. 4a, b).

3) In Figure 3H, it's unclear what the asterisks are indicating. Please clarify in figure legends and/or edit the plot to clearly indicate what comparisons are being made that the asterisks refer to.

We thank the Reviewer for highlighting this ambiguity. The plot is now edited and the figure legend provides additional information.

4) In Figure 5, the authors show that NPY KO eliminates the anxiolytic effect of hunger in the EOM assay, but it does not directly show that the anxiolytic effects of AgRP stimulation is mediated by NPY. In Figure 6, the authors nicely demonstrate that the synaptic effects of AgRP evoked NPY are absent in NPY KO mice, but this does not directly tie back to the

in vivo feeding results. The strongest evidence of this would be showing that AgRPBNST photostimulation in the EOM assay recapitulates how hunger increases feeding in the open areas and that this effect is absent in AgRP photostimulation in NPY-KO mice.

The Reviewer raises an important point that we are very thankful for. We followed the Reviewer's advice and conducted the experiment as suggested. As expected, we found that optogenetic stimulation of AgRP→BNST projections increases the time NPY-WT mice spent in the open arms during the EOM assay, but had no effect on the behavior of NPY-KO mice. These findings further support our hypothesis that the anxiolytic effect of AgRP neuron stimulation in the BNST requires AgRP neuron-mediated release of NPY. We include these additional data in our new Figure 5e-g.

5) The authors state that NPY-KO mice are not significantly different in the first 30 minutes of a re-feeding assay, but they do not acknowledge that they are significant at the 60-minute mark (Ext Fig 5). Furthermore, even in Extend Fig 5d, there does seem to be a reduction at 30 minutes but appears to be underpowered for statistical significance. I think this is important to address and discuss as the authors are using this as evidence of context-specific actions of NPY. To me it seems there are some global effects on feeding based on the data.

We thank the Reviewer for highlighting this important point. We now acknowledge the difference after 60 minutes of refeeding in the text (page 12, line 5 - 8). In addition, we performed additional feeding experiments in a new cohort of mice to increase the number of animals for the assessment of refeeding after fasting even further, in order to avoid any inaccurate conclusions due to underpowered statistical testing. Although our total group size is now 16 versus 14 animals, the acute increase in food intake after fasting is still not significantly different between NPY-WT and NPY-KO mice. These findings suggest that the lack of statistical significance is likely due to the high variability in feeding behavior of NPY-KO mice. In addition, we adapted our analysis to the first 20 minutes of refeeding (before: first 30 minutes), as this time period corresponds to the duration of the EOM assay. Our updated data are shown in the Extended Data Fig. 5c.

6) The statement that NPY does not affect neuronal activity of BNST neurons (Figure 6G) is suggestive and indirect. If these neurons are spontaneously active under cell-attached recording conditions, it would be better to record spontaneous firing under these less perturb conditions and show if NPY decreases firing rate of BNST neurons.

As suggested by the Reviewer, we performed additional electrophysiological experiments and recorded action-potential firing of BNST neurons in cell-attached mode. Our extensive experiments demonstrated that only a small fraction of BNST neurons is spontaneously active, which is in line with our whole-cell current-clamp experiments. Specifically, we found that only 3 out of 27 neurons that we recorded from reliably fired action potentials. Thus, the suggested experiments could only be conducted in a confined subset of BNST neurons, which might lead to an unintended selection bias. Consequently, we believe that our results presented in Fig. 6g properly demonstrate the effects of NPY signaling on BNST neurons.

We agree with the Reviewer that our original statement was indirect and we accordingly changed the wording to be more precise regarding our conclusions (page 14, line 23-24).

7) This discussion would benefit from a schematic/diagram of the model for how NPY influences the BNST circuits to impact anxiety and feeding would help. In particular, do the authors hypothesize that the increased synapse number of AgRP neurons onto BNST neurons actually matters for the anxiolytic effects observed or only an indirect consequence and that general extracellular release of NPY and action onto CeA presynaptic terminals is the actual mechanism? Can the authors discuss if there is expectation that CeA inputs would be near AgRP inputs or is it possible that diffusion of the peptide might be action on CeA inputs on totally different BNST neurons that the ones the AgRP neurons contact.

We thank the Reviewer for this very important comment and the stimulating suggestions of how to improve the discussion of our report. As suggested, we compiled a schematic summary of our findings and our drawn conclusions. In addition, we extended our discussion on the important aspects raised by the Reviewer. The schematic summary is shown in our new Extended Data Fig. 7 and the discussion points can be found on page 15, lines 12 - 19; page 16, lines 7 - 20.

Minor comments:

1) In the introduction, in the sentence "AgRP neuron activation decreases anxiety-related behavior and increases risk behavior to maximize food acquisition", could also cite Jikomes et al., 2016 DOI: 10.1016/j.cub.2016.07.019

We thank the Reviewer for the suggestion of this reference. We added the citation to the text (page 3, line 9).

2) Typo in methods section: "ad libidum" should be "ad libitum"

We thank the Reviewer for pointing out this (interesting) mistake, which is now corrected.

3) Can the authors clarify in the methods why 20 mW light was used for ARC experiments and whether the statement "rendering an irradiance of ~3-7 mW/mm²" was only for the BNST. This is unclear in two locations in the methods text whether the authors are saying 20 mW ARC stim was only leading to 3-7 mW of light reaching the ARC. Can authors calculate and state what irradiance for ARC would be given 20 mW was used?

We now include additional information on the irradiance in the BNST in the methods sections (page 26, lines 19 - 20).

4) Light On controls (i.e. mice receiving photostimulation but not expressing ChR2) would have been nice controls for CeA→BNST projections.

We performed additional experiments to address this important aspect. Our data on Light On controls for the optogenetic stimulation of CeA→BNST projections are now included in the Extended Data Fig. 1g.

5) Figure 2: please indicate in legends the number of slices in addition to the number of mice that the recordings came from. In other words, for the N=13 AgRP mice in 2C, how many slices were recorded from per mouse.

Information about the numbers of brain slices that we used for all of our electrophysiological experiments are now included in the figure legends.

6) A limitations of study section in the discussion would be nice and include whether CeA neurons collateralize and if this might contribute to the results. In addition, the choice of stimulation protocol as mentioned in major comment 1.

We thank the Reviewer for this suggestion and include a section in our discussion to acknowledge the limitations of our study.

REVIEWER COMMENTS

Reviewer #1 (Remarks to the Author):

The authors have largely well addressed the points I raised. I do remain concerned that much of the work is being performed in portions of the accumbens rather than the bed nucleus, and am unconvinced by the new images, which remain too zoomed in to be able to see additional neuroanatomical landmarks, but appear rostral to the bed nucleus.

Reviewer #2 (Remarks to the Author):

The authors have provided rationale and experimental support in addressing the reviews. This has been done thoroughly and entirely with the intent of improving the report. I have no concerns, and would accept the manuscript as is.

Reviewer #3 (Remarks to the Author):

The authors have completely addressed my concerns. I congratulate them on an interesting and well investigated study.

Response to Reviewers

„NPY-mediated synaptic plasticity in the extended amygdala prioritizes feeding during starvation”
(NCOMMS-23-40493-T)

We wish to thank the Reviewers for their positive response to our revised manuscript. As described below, we have performed additional histological analyses to directly address the remaining concern of Reviewer #1.

Reviewers' comments:

Reviewer #1:

The authors have largely well addressed the points I raised. I do remain concerned that much of the work is being performed in portions of the accumbens rather than the bed nucleus, and am unconvinced by the new images, which remain too zoomed in to be able to see additional neuroanatomical landmarks, but appear rostral to the bed nucleus.

We thank the Reviewer for acknowledging our extensive work to improve our manuscript.

*To resolve the remaining concern, we have performed additional microscopy of axonal projections and further extended our histological analyses. In particular, we now include additional representative images of the BNST and the nucleus accumbens. These images provide additional confirmation that AgRP neurons densely innervate the anterior part of the BNST, whereas there are almost no axonal terminals of AgRP neurons in the nucleus accumbens. Moreover, consistent with our previous analysis, GABAergic neurons of the CeA project to and densely innervate the BNST, including its anterior subdivision, but do not have axons innervating the nucleus accumbens. We include these additional images in our new **Extended Data Fig. 1**.*

Further, we now include numerous additional corresponding representative overview images of the BNST as well as the nucleus accumbens to complement our histological characterizations (Extended Data Fig. 3f, 5a, 5c). These new images include relevant anatomical landmarks, such as the third ventricle (3V), the lateral ventricle (LV), and the anterior commissure (ac), which allow the clear distinction between the BNST and the nucleus accumbens.

In addition, we would like to emphasize again that numerous previous publications have demonstrated that the BNST, but not the nucleus accumbens, is a major projection target of AgRP neurons (Betley et al., Cell (2013); Garfield et al., Nat Neurosci (2015); Steculorum et al., Cell (2016); Xia et al., Mol Psychiatry (2021)). These studies have also demonstrated that the optogenetic stimulation of AgRP→BNST projections potently promotes food consumption in sated

mice (Betley et al., Cell (2013); Garfield et al., Nat Neurosci (2015); Steculorum et al., Cell (2016)) and suppresses anxiety-related behavior (Xia et al., Mol Psychiatry (2021)). Importantly, to our knowledge, there are no comparable findings for the activation of AgRP neuron terminals in the nucleus accumbens.

Lastly, consistent with our findings, numerous previous studies investigating the projection profile of GABAergic CeA neurons have consistently demonstrated that the BNST, including its anterior subdivision, represents a major projection target. These studies have also shown that CeA neurons only sparsely innervate the nucleus accumbens with only very fibers, and do not list the nucleus accumbens as a primary target of the CeA (Alheid & Heimer, Neurosci (1988); Dong et al., Brain Res Rev (2001); Liu et al., Neurosci Lett (2021); Wang et al., eLife (2023)). Thus, our findings are entirely in line with previous reports.

Together, our extensive histological data confirm that all of our experiments were exclusively performed in the anterior part of the BNST and not in the nucleus accumbens. Our anatomical landmarks together with the known innervation patterns of AgRP and GABAergic CeA neurons allowed us to discriminate between the BNST and the nucleus accumbens in acute brain slices as well as histological sections.

We hope that our additional data now fully resolve the Reviewer's concern.

Reviewer #2:

The authors have provided rationale and experimental support in addressing the reviews. This has been done thoroughly and entirely with the intent of improving the report. I have no concerns, and would accept the manuscript as is.

We thank the Reviewer for the very positive response on our revised manuscript.

Reviewer #3:

The authors have completely addressed my concerns. I congratulate them on an interesting and well investigated study.

We thank the Reviewer for the very positive response on our revised manuscript.

REVIEWERS' COMMENTS

Reviewer #1 (Remarks to the Author):

The new S1 figure addresses the concern that I raised. I have no further concerns.